# Quantifying the Uncertainty of Foundation Models with Singular Value Ensembles

**Mehmet Ozgur Turkoglu** [1]   **Dominik J. Mühlematter** [2]   **Alexander Becker** [2]   **Konrad Schindler** [2]   **Helge Aasen** [1]

## Abstract

Foundation models have become a dominant paradigm in machine learning, achieving remarkable performance across diverse tasks through large-scale pretraining. However, they often yield overconfident, uncalibrated predictions. The standard approach to quantifying epistemic uncertainty are ensembles of multiple independently trained models. But their computational cost scales linearly with ensemble size, making them impractical for large foundation models. We propose Singular Value Ensemble (SVE), a parameter-efficient implicit ensembling method. SVE builds on a simple, but powerful core assumption: namely, that the singular vectors of the weight matrices correspond to meaningful directions in the representation space. If the singular vectors are indeed meaningful (orthogonal) "knowledge directions", then a model ensemble can be obtained by modulating only how strongly each direction contributes to the output. Rather than learning new parameters for each ensemble member, we freeze the singular vectors and only train per-member singular values that rescale the contribution of each direction in that shared knowledge basis. Ensemble diversity emerges naturally during joint training as stochastic initialization and random batch sampling cause different members to converge to different combinations of the same underlying knowledge. SVE performs comparable to an explicit ensemble, while increasing the parameter count of the base model by $<1\%$, making principled uncertainty estimation accessible in resource-constrained settings. We validate SVE on NLP and vision tasks with various different backbones and show that it improves calibration while maintaining predictive accuracy.

[1]Agroscope, Earth Observation of Agroecosystems, [2]ETH Zürich, Photogrammetry & Remote Sensing. Correspondence to: Mehmet Ozgur Turkoglu <moturkoglu@gmail.com>.

*Proceedings of the 43$^{rd}$ International Conference on Machine Learning*, Seoul, South Korea. PMLR 306, 2026. Copyright 2026 by the author(s).

## 1. Introduction

Machine learning models are increasingly deployed in high-stakes domains where incorrect predictions can have severe consequences, e.g., medical diagnosis, autonomous driving, financial risk assessment, and agricultural decision support. In such applications, a model must not only be accurate but also *know when it does not know*. A prediction delivered with undue confidence can be more dangerous than no prediction at all: an overconfident misdiagnosis may delay treatment, while an overconfident crop disease prediction may lead to unnecessary pesticide application or, worse, untreated outbreaks that devastate yields.

Uncertainty in predictions arises from two fundamentally different sources. *Aleatoric uncertainty* reflects irreducible noise inherent in the data, e.g., measurement error, label ambiguity, or stochastic processes. *Epistemic uncertainty*, in contrast, stems from the model's limited knowledge: regions of input space where training data is sparse, or where multiple hypotheses are consistent with observations. Epistemic uncertainty is reducible in principle; it quantifies what the model does not know and signals when predictions should not be trusted (Kendall & Gal, 2017).

Modern deep neural networks, despite their remarkable predictive power, are notoriously poor at quantifying their own uncertainty. Trained with maximum likelihood objectives, they tend to produce overconfident predictions, assigning near-certain probabilities even to inputs far from the training distribution (Guo et al., 2017). The dominant approach to epistemic uncertainty estimation remains to train an ensemble of independently initialized models (Lakshminarayanan et al., 2017). The principle is intuitive: near observed training samples, ensemble members agree; far from the data, they diverge. This disagreement serves as a proxy for epistemic uncertainty. Deep ensembles consistently outperform other Bayesian approximations such as MC-Dropout (Gal & Ghahramani, 2016), both in terms of calibration and for out-of-distribution detection.

However, deep ensembles suffer from a fundamental limitation: their computational cost scales linearly with the number of members. Maintaining an ensemble of $M$ independently trained models increases training cost and mem-

ory roughly linearly with $M$, regardless of model size. For foundation models with billions of parameters, even a modest ensemble with $M = 4$ is therefore often impractical. This cost particularly limits practitioners who rely on efficient fine-tuning: engineers adapting models with few GPUs, researchers deploying models on edge devices, and organizations without access to large compute clusters. In an era where parameter-efficient adaptation has made foundation models accessible to all, uncertainty quantification remains a privilege of the few.

The machine learning landscape has shifted rapidly towards foundation models as the default starting point. Pre-trained models continue to improve at a rapid pace, LLaMA, Mistral, and Qwen in language; DINOv3, and CLIP in vision, each generation offering stronger representations learned from ever-larger corpora. Concurrently, the community has embraced *parameter-efficient fine-tuning* (PEFT) as the default paradigm for adaptation: methods such as LoRA (Hu et al., 2022), adapters (Houlsby et al., 2019), and prompt tuning (Lester et al., 2021) enable practitioners to specialize billion-parameter models with modest hardware. As pre-training continues to improve, the representations encoded in these models become increasingly valuable, and the case for preserving them during fine-tuning grows stronger.

Recent advances in parameter-efficient fine-tuning offer a promising direction. A growing body of work has shown that the knowledge encoded in foundation models is organized along *linear subspaces*: the singular vectors of weight matrices correspond to semantically meaningful directions, while singular values encode their relative importance (Millidge & Black, 2022; Elhage et al., 2022). Aghajanyan et al. (2021) demonstrated that fine-tuning operates in a low-dimensional subspace, with the intrinsic dimension decreasing as models scale. Building on this insight, Sun et al. (2022) introduced Singular Value Fine-tuning (SVF), which decomposes pre-trained weights via SVD and trains *only the singular values* while freezing the singular vectors. Subsequent work has extended this paradigm: SVFit (Sun et al., 2024) scales singular value fine-tuning to large language models and vision transformers, manipulating $16\times$ fewer parameters than LoRA. These developments share a common insight: *the singular vectors learned during pre-training encode valuable knowledge that should be preserved, while singular values provide a compact parameterization for task-specific adaptation*. We hypothesize that this same principle can be leveraged to build model ensembles. If different rescalings of singular values yield functionally distinct models, then training per-member singular values should produce diverse ensemble members while sharing the same underlying knowledge basis.

We introduce Singular Value Ensemble (SV-Ensemble, SVE), a parameter-efficient implicit ensemble method that

implements this insight. SVE decomposes pre-trained weight matrices via Singular Value Decomposition, freezes the singular vectors (the shared knowledge basis), and trains only per-member singular values. Diversity among ensemble members emerges naturally from two sources: (i) stochastic initialization of singular values with small perturbations, and (ii) mini-batch sampling during joint training, which causes members to converge to distinct combinations of the same underlying subspaces.

This design yields extreme parameter efficiency. For a target weight matrix $W \in \mathbb{R}^{m \times n}$, low-rank adaptation methods add matrices with rank $r$ for each ensemble member (Mühlematter et al., 2026), requiring $\mathcal{O}((m + n)r)$ parameters per layer. In contrast, SVE adds only a vector of $\min(m, n)$ singular values per target matrix and per ensemble member. This makes SVE significantly more efficient than prior implicit ensemble methods, e.g., LoRA-Ensemble (Mühlematter et al., 2026), and orders of magnitude more efficient than explicit deep ensembles. SVE thus brings principled uncertainty quantification to the same resource-constrained settings where parameter-efficient fine-tuning has already democratized large, foundational models. Our contributions:

- We introduce Singular Value Ensemble (SVE), a parameter-efficient implicit ensemble method that creates diverse ensemble members by learning per-member singular values while sharing pre-trained singular vectors, the knowledge basis of a foundation model.

- We establish SVE as a new strong baseline for uncertainty quantification in foundation models, achieving state-of-the-art calibration while maintaining strong accuracy.

- We provide extensive validation across vision (Flowers102, CIFAR-100, DTD, Oxford Pets, OOD detection, CIFAR-100-C corruptions) and NLP (ARC-Easy, SST-2) benchmarks using DINOv1/v2, BERT, and LLaMA-2-7B backbones; thus demonstrating that SVE generalizes across model sizes ranging from 22M to 7B parameters.

**Conflict of Interest Disclosure.** The authors declare no financial conflicts of interest related to this work.

## 2. Related Work

**Estimation of Epistemic Uncertainty.** Quantifying epistemic uncertainty in neural networks remains challenging since exact posterior computation is intractable. Approximate Bayesian inference methods address this by placing priors on weights and approximating the posterior given training data. Variational inference approaches (Graves, 2011; Ranganath et al., 2014), notably Bayes by Backprop (Blundell et al., 2015), learn approximate posteriors through optimization. MCMC-based methods (Neal, 1996; Chen et al.,

2014), including SGLD (Welling & Teh, 2011), sample from the posterior directly. However, both families struggle with the high-dimensional, non-convex loss landscapes typical of deep learning (Gustafsson et al., 2020). Recent work adapts Bayesian inference to parameter-efficient fine-tuning: Yang et al. (2024) apply Laplace approximation to LoRA for improved calibration, while BLoB (Wang et al., 2024) jointly learns posterior mean and covariance during fine-tuning. More recently, C-LoRA (Rahmati et al., 2026) introduces a contextual Bayesian extension of LoRA, where lightweight data-dependent adapter modules dynamically modulate the posterior uncertainty for each input sample, while preserving the parameter efficiency of frozen-backbone adaptation. Alternative single-model approaches include SNGP (Liu et al., 2020), which combines spectral normalization with a Gaussian Process output layer for distance-aware uncertainty. However, SNGP does not transfer straightforwardly to transformer architectures and underperforms on sequence modeling tasks (Ulmer et al., 2022).

**Implicit Ensembling**   The computational complexity of deep ensembles (Lakshminarayanan et al., 2017) has motivated a line of work on *implicit ensembling* methods that emulate ensemble diversity without instantiating full separate models. MC-Dropout (Gal & Ghahramani, 2016) reinterprets dropout at inference as approximate Bayesian marginalization, but often underestimates uncertainty and produces overconfident predictions on out-of-distribution inputs. BatchEnsemble (Wen et al., 2020) introduces rank-one perturbations to shared weights, while MIMO (Havasi et al., 2021) trains multiple subnetworks within a single architecture. FiLM-Ensemble (Turkoglu et al., 2022) applies Feature-wise Linear Modulation to create implicit ensemble members through learned affine transformations of intermediate features, achieving strong calibration with reduced memory cost. More recently, LoRA-Ensemble (Mühlematter et al., 2026; Wang et al., 2023) extends Low-Rank Adaptation to implicit ensembling, training per-member low-rank matrices atop a frozen pretrained backbone. These methods substantially reduce parameter overhead but still require learning new parameters for each ensemble member, parameters that must learn useful directions in weight space from scratch.

**Low-Rank Adaptation.**   Parameter-efficient fine-tuning has been revolutionized by Low-Rank Adaptation (LoRA) (Hu et al., 2022), which injects trainable low-rank decomposition matrices into transformer layers while freezing pretrained weights. For a pretrained weight matrix $W_0 \in \mathbb{R}^{d \times k}$, LoRA parameterizes the update as $\Delta W = BA$, where $B \in \mathbb{R}^{d \times r}$, $A \in \mathbb{R}^{r \times k}$ with rank $r \ll \min(d, k)$, reducing trainable parameters by orders of magnitude. Several variants address LoRA's limitations: DoRA (Liu et al., 2024) decomposes weights into magnitude and direction

components, applying LoRA only to directional updates; AdaLoRA (Zhang et al., 2023) parameterizes updates directly in SVD form with adaptive rank allocation across layers; and VeRA (Kopiczko et al., 2024) achieves further efficiency by using frozen random low-rank matrices shared across layers with only trainable scaling vectors.

**Singular Value Fine-tuning.**   SVF (Sun et al., 2022) introduced the paradigm of decomposing pretrained weights via SVD ($W = U\Sigma V^\top$) and fine-tuning only the singular values $\Sigma$ while freezing the singular vectors. The key insight is that singular values act as a scaling mechanism that reweights representations without erasing semantic knowledge encoded in the singular vectors. Originally developed for few-shot segmentation, SVF achieved strong results with less than 1% of parameters trainable. SVFit (Sun et al., 2024) extends this approach to large language models and vision transformers, demonstrating that the top singular values capture nearly all of a matrix's information, justifying the focus on principal components. Experiments across RoBERTa, ViT, and Stable Diffusion show SVFit outperforms LoRA while requiring significantly fewer parameters. SVFT (Lingam et al., 2024) takes a complementary approach by parameterizing weight updates as sparse combinations of outer products of singular vectors ($\Delta W = UMV^\top$), enabling fine-grained control over expressivity with minimal parameters.

Several works couple SVD decomposition with LoRA's low-rank update structure. PiSSA (Meng et al., 2024) initializes LoRA adapters with principal components from SVD rather than random matrices, achieving faster convergence by updating the most important directions first. OSoRA (Han et al., 2025) unifies SVD initialization with learnable scaling, optimizing both singular values and output-dimension scaling vectors while keeping singular vectors frozen.

## 3. Background

**Foundation Models.**   Foundation models are large-scale neural networks pre-trained on massive datasets using self-supervised objectives, learning general-purpose representations that transfer to downstream tasks with minimal adaptation (Bommasani, 2021). In NLP, this paradigm emerged with BERT (Devlin et al., 2019) and the GPT family (Radford et al., 2019; Brown et al., 2020), with recent models like LLaMA (Touvron et al., 2023), Mistral (Jiang et al., 2023), and Qwen (Bai et al., 2023) advancing capabilities while improving efficiency. In vision, ViT (Dosovitskiy et al., 2021) demonstrated transformers can match convolutional networks, while self-supervised methods like DINO/DINOv2 (Caron et al., 2021; Oquab et al., 2024) and MAE (He et al., 2022) learn rich representations without labels. Multimodal models such as CLIP (Radford et al.,

2021) and SigLIP (Zhai et al., 2023) align vision-language representations through contrastive learning. A defining characteristic of foundation models is their ability to accumulate general, broadly applicable knowledge during pre-training. The model extracts statistical regularities, semantic relationships, and structural patterns from large-scale data. This knowledge is distributed across the model's weight matrices, which learn to map input signals to semantically meaningful representation spaces.

**Knowledge in Weight Space: Linear Subspaces.** Recent interpretability research has revealed that knowledge in transformers is organized along *linear subspaces* of weight matrices (Elhage et al., 2022; Millidge & Black, 2022). The SVD of a weight matrix $\mathbf{W} \in \mathbb{R}^{m \times n}$ decomposes it as:

$$\mathbf{W} = \mathbf{U}\boldsymbol{\Sigma}\mathbf{V}^\top = \sum_{i=1}^{r} \sigma_i \mathbf{u}_i \mathbf{v}_i^\top . \qquad (1)$$

Here, $\mathbf{U} \in \mathbb{R}^{m \times r}$ and $\mathbf{V} \in \mathbb{R}^{n \times r}$ contain left and right singular vectors forming an $r$-dimensional orthonormal basis ($r = \min(m, n)$), and $\boldsymbol{\Sigma} = \mathrm{diag}(\sigma_1, \ldots, \sigma_r)$ contains singular values in descending order. Empirical studies of transformer weights reveal key properties (Millidge & Black, 2022; Elhage et al., 2022; Staats et al., 2026):

- Singular vectors $\mathbf{u}_i$ and $\mathbf{v}_i$ correspond to semantically meaningful directions, yielding interpretable clusters when projected to embedding space (Millidge & Black, 2022).

- Singular value magnitudes quantify the importance of each direction, with both large and small values carrying meaningful information (Staats et al., 2026).

- Model adaptation can be achieved by manipulating specific SVD components, as shown by rank-one editing methods (Meng et al., 2022).

This suggests that singular vectors form a *knowledge basis*, orthogonal directions spanning the representation space, while singular values determine each direction's contribution. This motivates our approach: preserve the shared knowledge basis in singular vectors while learning only per-member singular values.

**Uncertainty Quantification via Ensembles.** Deep ensembles approximate the predictive posterior by averaging predictions from $M$ independently trained models:

$$p(y|\mathbf{x}) \approx \frac{1}{M} \sum_{m=1}^{M} p_{\theta_m}(y|\mathbf{x}) \qquad (2)$$

The diversity among ensemble members captures epistemic uncertainty, arising from limited or skewed training data. Implicit ensemble methods aim to achieve similar diversity with shared parameters.

## 4. Singular Value Ensemble

Our key insight is that by sharing the singular vectors (the "knowledge basis") across ensemble members while learning member-specific singular values, we can create functionally diverse predictions with minimal parameter overhead. Each ensemble member "reweights" the pretrained knowledge subspaces differently, leading to different predictions for the same input.

**SVD Parameterization.** For each target weight matrix $\mathbf{W} \in \mathbb{R}^{m \times n}$ of a pre-trained model, we compute its SVD:

$$\mathbf{W} = \mathbf{U}\boldsymbol{\Sigma}\mathbf{V}^\top \qquad (3)$$

We freeze the singular vectors $\mathbf{U}$ and $\mathbf{V}$, and replace the singular values $\boldsymbol{\Sigma}$ with $M$ trainable copies $\{\boldsymbol{\Sigma}^{(m)}\}_{m=1}^{M}$, one for each ensemble member. Each layer additionally maintains a per-member bias vector $\mathbf{b}^{(m)}$, initialized from the layer's pretrained bias. During inference, member $m$ transforms an input $\mathbf{x}$ as:

$$\mathbf{y}^{(m)} = \mathbf{W}^{(m)}\mathbf{x} + \mathbf{b}^{(m)}, \quad \mathbf{W}^{(m)} = \mathbf{U}\boldsymbol{\Sigma}^{(m)}\mathbf{V}^\top \qquad (4)$$

This parameterization combines anisotropic rescaling of the pretrained subspace (via $\boldsymbol{\Sigma}^{(m)}$) with a per-member translation (via $\mathbf{b}^{(m)}$), giving each member a slightly different mapping while preserving the shared singular-vector basis.

**Initialization and Diversity.** To encourage diversity while preserving pretrained structure, we employ multiplicative initialization. Each member's singular values $\boldsymbol{\Sigma}^{(m)}$ are initialized as:

$$\boldsymbol{\Sigma}^{(m)} = \boldsymbol{\Sigma} \odot (1 + \boldsymbol{\epsilon}^{(m)}), \quad \boldsymbol{\epsilon}^{(m)} \sim \mathcal{N}(\mathbf{0}, \sigma_{\text{init}}^2 \mathbf{I}) \qquad (5)$$

where $\odot$ denotes element-wise multiplication. This formulation scales perturbations proportionally to each singular value's magnitude, preserving relative ordering, and guarantees positivity for reasonable noise levels ($\sigma_{\text{init}} < 1$) at the beginning of training. Per-member biases are perturbed additively, $\mathbf{b}^{(m)} = \mathbf{b} + \boldsymbol{\eta}^{(m)}$ with $\boldsymbol{\eta}^{(m)} \sim \mathcal{N}(\mathbf{0}, \sigma_{\text{init}}^2 \mathbf{I})$. We empirically set $\sigma_{\text{init}} = 0.01$ in most experiments, corresponding to approximately 1% relative perturbation. Combined with stochastic mini-batch sampling during training, this small symmetry-breaking perturbation drives members toward distinct rescalings of the shared basis, analogous to how deep ensemble members diverge from different random initializations. Appendix E shows the resulting divergence of per-member singular values during training; Appendix C reports the sensitivity of $\sigma_{\text{init}}$.

**Implementation.** We apply SVE to all linear projection matrices in the transformer, including attention (query, key, value, output) and feed-forward layers; partial-layer variants are ablated in Appendix D. At initialization, we compute

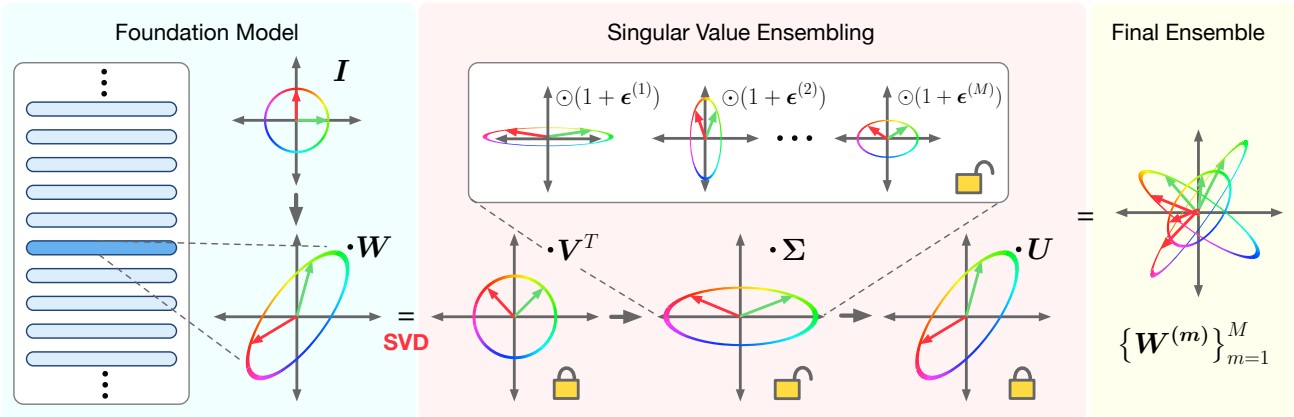

*Figure 1.* Schematic of Singular Value Ensembles (SV-Ensembles). *Left (blue background):* A given linear projection $\boldsymbol{W}$ of a Foundation Model visualized in 2D by means of its effect on the unit circle as well as two unit vectors. *Middle (red background):* $\boldsymbol{W}$ can be decomposed via SVD into orthogonal right-singular vectors $\boldsymbol{V}^T$, diagonal singular values $\boldsymbol{\Sigma}$, and orthogonal left-singular vectors $\boldsymbol{U}$. SV-Ensemble freezes $\boldsymbol{U}$ and $\boldsymbol{V}^T$ per layer and only learns $\boldsymbol{\Sigma}^{(m)}$ per ensemble member. *Right (yellow background):* Re-composing $\boldsymbol{W}^{(m)} = \boldsymbol{U}\boldsymbol{\Sigma}^{(m)}\boldsymbol{V}^T$ yields an ensemble of models with identical singular vectors that can be re-scaled per member.

the SVD of each target weight matrix, freeze the singular vectors $\mathbf{U}$ and $\mathbf{V}$, and instantiate $M$ trainable copies of the singular values $\boldsymbol{\Sigma}^{(m)}$ and bias vectors $\mathbf{b}^{(m)}$. At each forward pass, member $m$ reconstructs $\mathbf{W}^{(m)} = \mathbf{U}\boldsymbol{\Sigma}^{(m)}\mathbf{V}^\top$ with non-negativity of $\boldsymbol{\Sigma}^{(m)}$ enforced via `clamp(min=0)`. Each member maintains a separate classification head.[1]

**Training Objective.** All $M$ ensemble members are trained jointly with the average cross-entropy loss:

$$\mathcal{L} = \frac{1}{M} \sum_{m=1}^{M} \mathcal{L}_{\text{CE}}(f^{(m)}(\mathbf{x}), \mathbf{t}) \quad (6)$$

$f^{(m)}$ denotes the $m$-th ensemble member (parameterized by $\boldsymbol{\Sigma}^{(m)}$ and $\mathbf{b}^{(m)}$) and $\mathbf{t} \in \{0,1\}^C$ is the one-hot label.

**Parameter Efficiency.** Per ensemble member, SVE adds $\min(m,n)$ trainable singular values per weight matrix, plus $m$ bias entries when the layer has a bias. For a transformer with $L$ layers and hidden dimension $d$, applying SVE to the four attention projections (each $d \times d$) and the two MLP layers ($d \times 4d$ and $4d \times d$) contributes $6d$ singular values and $9d$ bias entries per layer per member. The total overhead relative to the base model weights is:

$$\text{Overhead} = \frac{(M-1) \cdot L \cdot (6d + 9d)}{L \cdot 12d^2} = \frac{5(M-1)}{4d} \lesssim 1\% \quad (7)$$

for $d = 768$ and $M \leq 8$. For bias-free architectures (e.g., LLaMA), the bias term vanishes, and the overhead reduces to $(M-1)/(2d)$, which is even smaller, only $\approx 0.2\%$ for LLaMA-2-7B ($d = 4096$) at $M = 16$. We additionally

allocate a separate classification head per ensemble member, adding $M \cdot d \cdot C$ parameters for $C$ classes, which remains negligible for most applications. Further parameter efficiency can be achieved by adapting only a subset of the weight matrices; see Appendix D.

## 5. Experiments

We evaluate on diverse tasks spanning both NLP and vision domains, deliberately selecting datasets that reflect practical deployment scenarios where uncertainty quantification matters most: moderate-scale problems where practitioners fine-tune foundation models on domain-specific data with limited training samples. For NLP: ARC-Easy (Clark et al., 2018) (multiple-choice question answering; 2,251 training / 2,376 test samples) and SST-2 (Socher et al., 2013) (binary sentiment classification; 67k training / 872 validation samples). For vision: Flowers102 (Nilsback & Zisserman, 2008) (102 flower classes; 1,020 training / 6,149 test), CIFAR-100 (Krizhevsky, 2009) (100 object classes; 50k training / 10k test), DTD (Cimpoi et al., 2014) (47 texture classes; 1,880 training / 1,880 test), and Oxford Pets (Parkhi et al., 2012) (37 pet breeds; 3,680 training / 3,669 test). These datasets represent realistic fine-tuning regimes: fine-grained classification with limited labels (Flowers102, DTD, Oxford Pets), general object recognition (CIFAR-100), sentiment analysis (SST-2), and commonsense reasoning (ARC-Easy). For robustness evaluation, we use CIFAR-10 (10 classes; 10k test) as an out-of-distribution dataset and CIFAR-100-C (Hendrycks & Dietterich, 2019) for distribution shift experiments with five severity levels. Refer to Appendix G for more details about the datasets.

We compare against the most established and widely-adopted uncertainty quantification methods for neural net-

---

[1] Our implementation is available at https://github.com/moturkoglu/Singular-Value-Ensemble.

works. As reference points, we include Single Model (standard fine-tuning) and Single w/ SVF (singular value fine-tuning without ensembling). Deep Ensemble (Lakshminarayanan et al., 2017) serves as the gold standard for uncertainty estimation, consistently ranking among the top methods in comprehensive benchmarks. MC Dropout (Gal & Ghahramani, 2016) remains one of the most widely deployed approaches due to its simplicity and integration into major frameworks. For implicit ensembles, we compare against Batch Ensemble (Wen et al., 2020), included in Google's Uncertainty Baselines library, and LoRA-Ensemble (Mühlematter et al., 2026; Wang et al., 2023), the current state-of-the-art for implicit ensembling of transformer architectures. Finally, we include recent Bayesian and related approaches designed for efficient fine-tuning: Bayes-LoRA (Yang et al., 2024), which applies a Laplace approximation over LoRA parameters, and BLoB (Wang et al., 2024), which jointly learns the posterior mean and covariance during fine-tuning and represents a state-of-the-art Bayesian approach for LLMs. We additionally include C-LoRA (Rahmati et al., 2026) on ARC-Easy as a recent contextual Bayesian LoRA method for uncertainty estimation, and Evidential Deep Learning (EDL) (Sensoy et al., 2018) on Flowers102 as an additional single-model uncertainty baseline.

We evaluated each method's predictive performance using classification accuracy and its calibration quality through Expected Calibration Error (ECE), Negative Log-Likelihood (NLL), and Brier score. The ECE measures how far predicted confidence intervals deviate from observed error rates; perfect calibration occurs when the estimated uncertainties match the actual likelihood of misclassification. Definitions of all metrics are provided in Appendix H.

For NLP, we use LLaMA-2-7B for ARC-Easy and BERT-base-uncased for SST-2. For vision, we use DINO ViT-S/16 and DINOv2 ViT-S/14. All experiments use $M = 4$ ensemble members for vision tasks, $M = 8$ for SST-2 and $M = 16$ for ARC-Easy. Refer to Appendix B for how performance changes with a different number of members. Refer to Appendix F for training details.

**Effect of Backbone Pretraining Scale.** We first investigate how the quality of pretrained representations affects ensemble performance. Figure 2 shows CIFAR-100 test performance across progressively stronger pretrained backbones: random initialization, DINOv1, and DINOv2. A striking pattern emerges: as backbone quality improves because of larger pre-training dataset sizes, SV-Ensemble shows increasingly larger gains relative to other ensemble baselines. With randomly initialized weights, SV-Ensemble performs the worst as there is no meaningful knowledge base to leverage, while a single or deep ensemble can converge to a better solution. However, with DINOv2's superior

representations, SV-Ensemble achieves the largest improvement and outperforms all the baselines, including Deep Ensemble. This validates our core hypothesis: the benefits of SVE grow with representation quality, as stronger pretraining produces more meaningful singular vector bases that can be effectively reweighted for ensemble diversity.

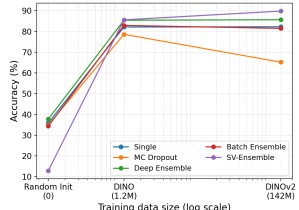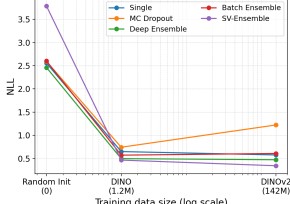

*Figure 2.* CIFAR-100 test performance vs. pretraining scale for a ViT-S model. With progressively stronger backbones (Random Init, DINOv1, DINOv2), SV-Ensembles show increasingly larger gains relative to other ensembling schemes. I.e., the benefits of SV-based ensembling grow with representation quality.

**Vision Results.** Table 1 presents results for four vision benchmarks, using DINO backbones with 4 ensemble members. Models are trained for 10 epochs except Oxford Pets, where we train for only ≈200 iterations to evaluate convergence speed and mimic resource-constrained training.

Across all experiments, MC Dropout consistently falls short in terms of accuracy, often performing worse than even the single model baseline. While it occasionally provides marginally better calibration than the single model, it fails to leverage the pretrained representations effectively. This observation aligns with prior findings that MC Dropout is not well-suited for pretrained transformer architecture (Mühlematter et al., 2026). Evidential Deep Learning (EDL) also underperforms on Flowers102, lowering accuracy and severely hurting calibration compared to a single model. Temperature scaling mostly fixes calibration, but accuracy still stays much lower. Single w/ SVF already improves both accuracy and calibration over the single model across all datasets, validating the effectiveness of singular value fine-tuning as a standalone adaptation strategy. Batch Ensemble shows mixed results: it achieves reasonable accuracy and calibration on some datasets (Flowers102, CIFAR-100), but catastrophically fails on Oxford Pets with severely degraded calibration. This instability suggests that Batch Ensemble's rank-one perturbations are not well-suited for constrained training regimes with limited iterations. Deep Ensemble consistently improves accuracy over the single model and provides strong calibration on most datasets, except Oxford Pets where it exhibits poor calibration despite achieving good accuracy. LoRA-Ensemble achieves the best results on CIFAR-100, the largest dataset with 100 classes, outperforming all other methods in both accuracy and calibration. However, on the remaining three datasets,

*Table 1.* Comparison on vision benchmarks using DINO back-bones. 4 members are used for ensemble methods. For each metric the best score is in **bold**, second best underlined.

| Method | Acc (%)↑ | ECE (%)↓ | NLL↓ | Brier↓ |
|---|---|---|---|---|
| *Flowers102 – DINO ViT-S/16 – Fine-tuning for 10 epochs* | | | | |
| Single | $86.3_{\pm0.8}$ | $3.9_{\pm0.8}$ | $0.56_{\pm0.05}$ | $0.20_{\pm0.02}$ |
| EDL | 80.4 | 71.9 | 2.85 | 0.85 |
| EDL+Temp. Scaling | 80.4 | 4.8 | 0.89 | 0.26 |
| Single w/ SVF | $91.8_{\pm0.4}$ | $1.8_{\pm0.4}$ | $0.33_{\pm0.01}$ | $0.12_{\pm0.01}$ |
| MC Dropout | $76.6_{\pm2.5}$ | $4.4_{\pm0.3}$ | $0.92_{\pm0.01}$ | $0.32_{\pm0.03}$ |
| Deep Ensemble | $91.5_{\pm0.3}$ | $\mathbf{0.9}_{\pm0.2}$ | $0.33_{\pm0.01}$ | $0.12_{\pm0.00}$ |
| Batch Ensemble | $91.7_{\pm0.4}$ | $2.2_{\pm0.2}$ | $0.32_{\pm0.02}$ | $0.12_{\pm0.01}$ |
| LoRA-Ensemble | $\underline{94.6}_{\pm0.1}$ | $1.1_{\pm0.1}$ | $\underline{0.21}_{\pm0.01}$ | $\underline{0.08}_{\pm0.00}$ |
| SV-Ensemble | $\mathbf{95.4}_{\pm0.2}$ | $\underline{1.0}_{\pm0.1}$ | $\mathbf{0.18}_{\pm0.01}$ | $\mathbf{0.07}_{\pm0.00}$ |
| *CIFAR100 – DINO ViT-S/16 – Fine-tuning for 10 epochs* | | | | |
| Single | $82.1_{\pm0.1}$ | $6.2_{\pm0.3}$ | $0.65_{\pm0.01}$ | $0.26_{\pm0.00}$ |
| Single w/ SVF | $82.6_{\pm0.1}$ | $\mathbf{1.1}_{\pm0.1}$ | $0.57_{\pm0.01}$ | $0.25_{\pm0.00}$ |
| MC Dropout | $78.6_{\pm0.7}$ | $5.3_{\pm0.5}$ | $0.74_{\pm0.02}$ | $0.30_{\pm0.01}$ |
| Deep Ensemble | $85.4_{\pm0.1}$ | $3.9_{\pm0.1}$ | $0.50_{\pm0.00}$ | $\underline{0.21}_{\pm0.00}$ |
| Batch Ensemble | $82.9_{\pm0.5}$ | $2.9_{\pm0.3}$ | $0.57_{\pm0.01}$ | $0.24_{\pm0.01}$ |
| LoRA-Ensemble | $\mathbf{88.2}_{\pm0.1}$ | $1.1_{\pm0.2}$ | $\mathbf{0.37}_{\pm0.00}$ | $\mathbf{0.17}_{\pm0.00}$ |
| SV-Ensemble | $\underline{85.6}_{\pm0.2}$ | $2.1_{\pm0.2}$ | $\underline{0.47}_{\pm0.00}$ | $\underline{0.21}_{\pm0.00}$ |
| *DTD Texture – DINOv2 ViT-S/14 – Fine-tuning for 10 epochs* | | | | |
| Single | $60.2_{\pm0.9}$ | $11.6_{\pm1.4}$ | $1.56_{\pm0.05}$ | $0.55_{\pm0.02}$ |
| Single w/ SVF | $\underline{78.4}_{\pm0.9}$ | $8.2_{\pm0.6}$ | $0.82_{\pm0.04}$ | $\underline{0.32}_{\pm0.01}$ |
| MC Dropout | $53.1_{\pm2.8}$ | $9.2_{\pm2.1}$ | $1.76_{\pm0.07}$ | $0.62_{\pm0.04}$ |
| Deep Ensemble | $65.4_{\pm0.6}$ | $7.6_{\pm0.6}$ | $1.32_{\pm0.01}$ | $0.48_{\pm0.00}$ |
| Batch Ensemble | $67.2_{\pm0.5}$ | $8.1_{\pm0.4}$ | $1.20_{\pm0.01}$ | $0.47_{\pm0.01}$ |
| LoRA-Ensemble | $\underline{78.4}_{\pm0.7}$ | $\underline{6.6}_{\pm0.9}$ | $\underline{0.80}_{\pm0.01}$ | $\underline{0.32}_{\pm0.01}$ |
| SV-Ensemble | $\mathbf{79.5}_{\pm0.1}$ | $\mathbf{6.3}_{\pm0.2}$ | $\mathbf{0.74}_{\pm0.01}$ | $\mathbf{0.30}_{\pm0.01}$ |
| *Oxford Pets – DINOv2 ViT-S/14 – Fine-tuning for ≈ 200 iter.* | | | | |
| Single | $81.4_{\pm1.1}$ | $8.6_{\pm0.8}$ | $0.61_{\pm0.03}$ | $0.27_{\pm0.01}$ |
| Single w/ SVF | $87.2_{\pm1.4}$ | $\mathbf{1.9}_{\pm0.8}$ | $\underline{0.40}_{\pm0.06}$ | $0.19_{\pm0.02}$ |
| MC Dropout | $25.9_{\pm15.7}$ | $6.1_{\pm2.4}$ | $2.53_{\pm0.63}$ | $0.85_{\pm0.12}$ |
| Deep Ensemble | $\underline{89.2}_{\pm0.4}$ | $13.3_{\pm0.5}$ | $0.43_{\pm0.02}$ | $\underline{0.17}_{\pm0.04}$ |
| Batch Ensemble | $70.5_{\pm2.4}$ | $48.7_{\pm1.7}$ | $1.75_{\pm0.05}$ | $0.69_{\pm0.02}$ |
| LoRA-Ensemble | $86.1_{\pm1.0}$ | $9.0_{\pm4.2}$ | $0.49_{\pm0.06}$ | $0.22_{\pm0.02}$ |
| SV-Ensemble | $\mathbf{90.1}_{\pm1.3}$ | $\underline{2.2}_{\pm0.9}$ | $\mathbf{0.30}_{\pm0.03}$ | $\mathbf{0.15}_{\pm0.02}$ |

SV-Ensemble achieves the best accuracy and competitive or superior calibration. On Oxford Pets, SV-Ensemble achieves the highest accuracy and calibration, substantially outperforming Deep Ensemble and LoRA-Ensemble, which both suffer from poor ECE in this low-iteration setting. Overall, SV-Ensemble demonstrates robust performance across diverse datasets and training regimes, achieving the best or second-best results in most settings while using an order of magnitude fewer parameters than LoRA-Ensemble.

**NLP Results.** Table 2 presents results on SST-2 sentiment classification using BERT-base with 8 ensemble members, trained for 3 epochs. SV-Ensemble achieves the best calibration among all methods, improving substantially over Deep Ensemble, LoRA-Ensemble, and Bayes-LoRA. While Deep Ensemble achieves the highest accuracy, SV-Ensemble remains competitive, trailing only slightly behind LoRA-Ensemble while using an order of magnitude fewer parameters. MC Dropout performs poorly on this task too, consistent with vision experiments and literature that dropout-

based uncertainty tends to underperform on transformer architectures (Mühlematter et al., 2026).

Table 3 presents results on ARC-Easy reasoning using LLaMA-2-7B. This challenging benchmark reveals how different approaches trade off accuracy and calibration. In terms of accuracy, LoRA-Ensemble, Deep Ensemble, and Checkpoint Ensemble all perform similarly, with SV-Ensemble matching this level at $M = 16$ members (for results with a different number of members, refer to Appendix B). The key differentiator is calibration: SV-Ensemble achieves dramatically lower ECE than all explicit and implicit ensemble baselines. Notably, SV-Ensemble approaches the calibration quality of Blob, a Bayesian method that jointly learns mean and covariance during fine-tuning, while being substantially simpler to implement. This positions SV-Ensemble as a balanced approach: it achieves accuracy comparable to LoRA-Ensemble and calibration comparable to Bayesian methods, while being more computationally efficient (see Table 5). The diversity induced by per-member singular values captures meaningful epistemic uncertainty without requiring explicit posterior approximation or expensive sampling procedures.

*Table 2.* Performance on the SST-2 validation dataset. Ensembles have 8 members. Best score for each metric in **bold**, second-best underlined.

| Method | Acc(%) (↑) | ECE (%) (↓) | NLL (↓) |
|---|---|---|---|
| Single | $92.5_{\pm0.2}$ | $6.4_{\pm0.3}$ | $0.35_{\pm0.01}$ |
| Single w/ LoRA | $91.6_{\pm0.5}$ | $5.9_{\pm0.5}$ | $0.29_{\pm0.02}$ |
| Single w/ SVF | $91.5_{\pm0.2}$ | $4.0_{\pm0.3}$ | $0.24_{\pm0.01}$ |
| MC Dropout | $84.9_{\pm1.2}$ | $6.1_{\pm0.4}$ | $0.36_{\pm0.02}$ |
| Deep Ensemble | $\mathbf{93.2}_{\pm0.2}$ | $4.7_{\pm0.2}$ | $\underline{0.23}_{\pm0.00}$ |
| LoRA-Ensemble | $\underline{92.7}_{\pm0.2}$ | $3.8_{\pm0.3}$ | $\mathbf{0.21}_{\pm0.01}$ |
| Bayes-LoRA | $90.7_{\pm0.3}$ | $\underline{3.6}_{\pm0.3}$ | $0.25_{\pm0.01}$ |
| SV-Ensemble | $92.0_{\pm0.1}$ | $\mathbf{2.8}_{\pm0.2}$ | $\mathbf{0.21}_{\pm0.01}$ |

*Table 3.* Comparison for the ARC-Easy (multiple choice QA) reasoning task with LLaMA-2-7B. Best score for each metric in **bold**, second-best underlined.

| Method | Acc (%) ↑ | ECE (%) ↓ | NLL ↓ |
|---|---|---|---|
| Single | 84.7 | 13.4 | 1.26 |
| Single w/ SVF | 83.2 | 12.1 | 0.77 |
| MC Dropout (N=10) (Yang et al., 2024) | 85.0 | 12.4 | 1.11 |
| Deep Ensemble (M=3) (Yang et al., 2024) | $\underline{85.8}$ | 9.9 | 0.83 |
| Last-Layer Ensemble (M=5) (Wang et al., 2023) | 72.0 | 6.0 | 0.79 |
| Ckpt Ensemble (M=3) (Yang et al., 2024) | $\underline{85.8}$ | 9.8 | 0.80 |
| LoRA-Ensemble (M=5) (Wang et al., 2023) | $\mathbf{86.0}$ | 9.0 | 0.92 |
| C-LoRA (Rahmati et al., 2026) | 84.4 | 4.3 | 0.48 |
| Bayes-LoRA (LLLA) (Yang et al., 2024) | 84.7 | 11.6 | 0.87 |
| Bayes-LoRA (LA) (Yang et al., 2024) | 85.1 | 5.4 | 0.49 |
| Blob (N=5) (Wang et al., 2024) | 84.7 | 6.2 | 0.46 |
| Blob (N=10) (Wang et al., 2024) | 85.5 | $\mathbf{3.6}$ | $\mathbf{0.40}$ |
| SV-Ensemble (M=16) | $\underline{85.8}$ | $\underline{3.8}$ | $\underline{0.43}$ |

**OOD Detection & Dataset Shift Robustness.** We evaluate SVE under both out-of-distribution (OOD) inputs and

*Table 4.* Model performance on out-of-distribution (OOD) detection. CIFAR-100 (DINO ViT-S/14 fine-tuned for 10 epochs) is used as the in-distribution dataset, with CIFAR-10 as OOD dataset. All ensemble methods use 4 members. For each metric, the best score is shown in **bold** and the second-best is underlined.

| Method | AUROC (↑) | AUPRC (↑) | FPR 95% TPR (↓) |
|---|---|---|---|
| | CIFAR-100 → CIFAR-10 (OOD) | | |
| Single | $76.6_{\pm 0.4}$ | $79.8_{\pm 0.5}$ | $80.9_{\pm 0.2}$ |
| Single w/ SVF | $77.6_{\pm 1.1}$ | $79.5_{\pm 1.5}$ | $75.3_{\pm 0.9}$ |
| MC Dropout | $75.0_{\pm 0.1}$ | $77.9_{\pm 0.3}$ | $81.7_{\pm 0.4}$ |
| Deep Ensemble | $79.2_{\pm 0.2}$ | $82.2_{\pm 0.2}$ | $78.0_{\pm 0.5}$ |
| Batch Ensemble | $78.8_{\pm 0.1}$ | $81.4_{\pm 0.2}$ | $75.8_{\pm 0.7}$ |
| LoRA-Ensemble | $\mathbf{82.8}_{\pm 0.4}$ | $\mathbf{84.4}_{\pm 0.5}$ | $\mathbf{64.3}_{\pm 0.6}$ |
| SV-Ensemble | $\underline{81.6}_{\pm 0.3}$ | $\underline{83.0}_{\pm 0.5}$ | $\underline{66.8}_{\pm 0.2}$ |

controlled dataset shifts, two key challenges for reliable uncertainty estimation in deep learning (Hendrycks & Gimpel, 2017). For OOD detection, models are trained on CIFAR-100 and evaluated on both CIFAR-100 (in-distribution) and CIFAR-10 (OOD), using the maximum softmax probability as the confidence score following prior work (Sim et al., 2023; Chen et al., 2024). As shown in Table 4, SV-Ensemble achieves strong OOD detection performance across all metrics, ranking second overall behind LoRA-Ensemble while using substantially fewer parameters.

To assess robustness under dataset shift, we further evaluate SV-Ensemble on CIFAR-100-C, which apply 19 corruption types at five severity levels to the original test sets (Hendrycks & Dietterich, 2019). Across corruption severities, SV-Ensemble matches the accuracy of state-of-the-art implicit ensembles despite using far fewer parameters. More importantly, it yields superior calibration, especially at higher severities, surpassing even Deep Ensemble. SV-Ensemble also maintains a strong NLL, remaining comparable to LoRA-Ensemble under stronger corruptions.

**Epistemic Uncertainty as Mutual Information.** The metrics above (calibration, NLL, OOD detection, and shift robustness) already indicate that SVE captures useful uncertainty. To probe the epistemic component directly, we use the standard decomposition of predictive uncertainty (Depeweg et al., 2018; Hüllermeier et al., 2022): the total uncertainty, given by the predictive entropy of the averaged prediction, is decomposed into two terms: the aleatoric part, corresponding to the mean per-member entropy; and the epistemic part, corresponding to the mutual information $\mathrm{MI} = H[\bar{p}] - \frac{1}{M}\sum_m H[p_m]$. Intuitively, MI is large only when ensemble members are individually confident but mutually inconsistent, that is, when the model finds multiple different answers likely and is uncertain which of them is correct. An analysis on CIFAR-100 (ID) versus CIFAR-10 (OOD) shows that SVE's restricted parameterization preserves this signal, see Appendix A and Table 6.

*Table 5.* Computational cost comparison (relative to single network) of major competing approaches using Bert-base on the SST2 task ($M = 8$ ensemble members). A batch size of 1 and a sequence length of 128 are used for timings. Best score is in **bold**.

| Method | Parameter Overhead(%) | Memory Overhead(%) | Infer. Flops(G) | Infer. Time(ms) |
|---|---|---|---|---|
| Deep Ensemble | 700 | 700 | $8 \times 21.9$ | $8 \times 3.1$ |
| LoRA-Ensemble | 10.3 | 10.3 | 175.2 | **21.9** |
| Bayes-LoRA | 4.0 | 4.0 | **68.2** | 313 |
| SV-Ensemble | **1.0** | **1.0** | 175.2 | **21.9** |

The OOD/ID ratio is higher ($3.4\times$ vs. $3.2\times$) in SVE than in an explicit Deep Ensemble, despite its MI being much lower in absolute terms. I.e., disagreement between members is low for in-distribution inputs and rises sharply under distribution shift. This concentration is not overconfidence on in-distribution data: in selective classification on CIFAR-100 (Table 7), SVE rejects misclassified samples at essentially the same rate as the Deep Ensemble. This behavior is consistent with the loss-landscape view of Fort et al. (2019), where ensemble diversity arises from exploring a structured, low-dimensional subspace. SVE makes the subspace explicit by constraining members to the lower-dimensional space spanned by the pretrained singular vectors, but is still able to explore that space.

**Computational Cost.** Table 5 compares computational overhead on SST-2 using BERT-base with $M = 8$ members (RTX 4090, batch size 1, sequence length 128). SV-Ensemble achieves the lowest parameter and memory overhead at just 1.0%, representing a $10\times$ reduction over LoRA-Ensemble and $700\times$ over Deep Ensemble. For inference, SV-Ensemble matches LoRA-Ensemble as the fastest method. Bayes-LoRA, while parameter-efficient, incurs prohibitive inference latency due to posterior sampling. The SVD decomposition required by SV-Ensemble is a one-time initialization cost (approximately 3.5 seconds for BERT-base), negligible relative to total training time. Note that our current implementation parallelizes ensemble members on a single GPU, so memory savings are not fully realized at runtime. In memory-constrained setups, members can be processed sequentially during training and inference. We leave optimized sequential implementations to future work.

## 6. Conclusion

We presented SV-Ensemble, a parameter-efficient method for uncertainty quantification in foundation models. By decomposing pretrained weights via SVD and training only per-member singular values while sharing singular vectors, our approach creates diverse ensemble members with minimal overhead. Our experiments across NLP, vision, OOD detection, and robustness benchmarks demonstrate that SV-Ensemble achieves calibration comparable to or superior

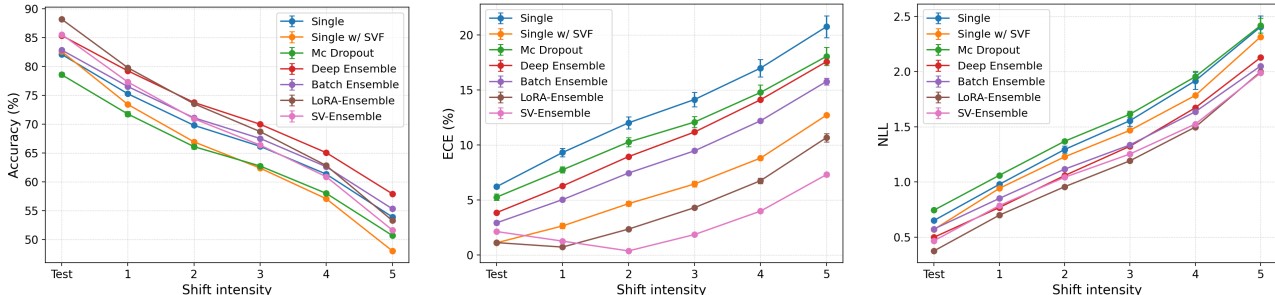

*Figure 3.* Dataset Robustness Comparison with varying shift intensity on CIFAR-100-C. (i) Accuracy, (ii) Expected Calibration Error (ECE), and (iii) Negative Log-Likelihood (NLL).

to that of deep ensembles and existing implicit ensemble methods, while being significantly more parameter-efficient. Validated across model scales with 22M to 7B parameters, we establish SV-Ensemble as a strong baseline for uncertainty quantification in foundation models, particularly suited to resource-constrained applications that require both reliable uncertainty estimates and computational efficiency. As foundation models continue to improve, strong priors from large-scale pretraining can solve diverse downstream tasks with minimal adaptation. SV-Ensemble leverages this by preserving representations encoded in shared singular vectors while introducing diversity through lightweight singular value adjustments.

**Limitations.** Although SVE substantially reduces the number of trainable parameters, inference still requires one forward pass per ensemble member, so its FLOPs remain comparable to deep ensembles; this is a general property of implicit ensemble methods rather than a limitation specific to SVE. Integration with 4/8-bit-quantized backbones is also not straightforward, since dequantization is required before computing the SVD. Finally, because SVE preserves the pretrained singular-vector basis and only rescales existing directions, it can inherit biases or spurious correlations encoded in the backbone. Our evaluation covers a limited set of models and datasets, and effectiveness may vary across other architectures and tasks.

**Future Work.** A promising direction is compressing implicit ensembles into a single network via knowledge distillation, preserving calibrated uncertainty estimates while reducing inference time and memory to that of a single forward pass. Another avenue worth exploring is combining SV-Ensemble with LoRA-Ensemble, since both methods achieve top performance in complementary settings: SV-Ensemble excels in calibration and parameter efficiency, while LoRA-Ensemble provides additional expressivity for complex tasks. A hybrid approach that leverages singular value diversity alongside low-rank adaptation could potentially capture the benefits of both methods.

## Impact Statement

Efficient, implicit ensembling methods like SV-Ensemble reduce parameter- and memory overhead required when training model ensembles for uncertainty estimation as well as mitigation of overconfident predictions. Beyond accessibility (ensembles can now be trained or deployed in scenarios where full ensembling is infeasible), this also has practical benefits: efficient ensembling has the potential to drastically reduce memory usage and energy consumption, especially when applied to large foundation models, contributing to more sustainable use of computational resources.

At the same time, uncertainty estimates should not be interpreted as guarantees of safety or correctness. Domain shifts between training and deployment conditions can lead to degradation of uncertainty estimation capabilities, and responsible use therefore requires validation in the target setting and appropriate human oversight.

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

# A. Additional Epistemic Uncertainty Evaluation

This appendix expands on the epistemic uncertainty analysis in the main text. For OOD detection in the main experiments (Table 4), we follow standard practice and score each input by the maximum softmax probability (MSP) of the averaged ensemble prediction, $\max_y \bar{p}(y \mid x)$, where $\bar{p}(y \mid x) = \frac{1}{M} \sum_{m=1}^{M} p_m(y \mid x)$. To isolate the epistemic component, we additionally consider the predictive entropy $H[\bar{p}(y \mid x)]$, which captures total uncertainty, and the mutual information

$$\mathrm{MI}(x) = H[\bar{p}(y \mid x)] - \frac{1}{M} \sum_{m=1}^{M} H[p_m(y \mid x)],$$

which captures the epistemic part alone, i.e., the disagreement between ensemble members. The results in this section come from a single run. They are meant to illustrate the relative behavior of these scores, but are not directly comparable to the multi-seed averages in Table 4.

**OOD Detection.**   Table 6 reports OOD detection results for CIFAR-100 (ID) versus CIFAR-10 (OOD) using entropy and MI scores, together with the mean uncertainty assigned to ID and OOD inputs and their ratio. SVE results in a lower absolute MI than the Deep Ensemble, yet its OOD/ID ratio is higher (3.4× vs 3.2×): members agree closely on in-distribution inputs and diverge sharply only under distribution shift, so the epistemic signal is concentrated where it is informative. The Deep Ensemble, whose members are trained independently, produces larger but a bit more diffuse disagreements, i.e., they tend to be spread across both ID and OOD inputs.

*Table 6.* OOD detection on CIFAR-100 (ID) vs. CIFAR-10 (OOD) using predictive entropy and mutual information (MI). ID mean and OOD mean are the average uncertainty on each split, and OOD/ID is their ratio. For each column, the better of the two methods is shown in **bold** (except descriptive ID/OOD means).

| Score | Method | AUROC(↑) | AUPRC(↑) | FPR 95% TPR(↓) | ID mean | OOD mean | OOD/ID |
|---|---|---|---|---|---|---|---|
| Entropy | Deep Ens. | 81.9 | 79.1 | 56.7 | 0.451 | 1.247 | 2.8× |
| Entropy | SVE | **83.9** | **83.3** | **56.5** | 0.578 | 1.728 | **3.0×** |
| MI | Deep Ens. | 82.5 | 79.3 | **57.0** | 0.110 | 0.354 | 3.2× |
| MI | SVE | **82.7** | **80.6** | 58.6 | 0.007 | 0.026 | **3.4×** |

**Selective Classification.**   We further evaluate the usefulness of the uncertainty estimates through selective classification on the CIFAR-100 test set: samples are ranked by uncertainty and the most uncertain predictions are rejected, so a meaningful score should yield higher accuracy as coverage decreases. We report the raccuracy when retaining 90%, 80%, and 70% coverage, and the area under the risk-coverage curve (AURC, lower is better).

*Table 7.* Selective classification on CIFAR-100 ID test. For each column, the better of the two methods is shown in **bold**.

| Score | Method | Base(↑) | @90%(↑) | @80%(↑) | @70%(↑) | AURC(↓) |
|---|---|---|---|---|---|---|
| MSP | Deep Ens. | 85.1 | 90.3 | 94.2 | **97.1** | 0.028 |
| MSP | SVE | **85.8** | **90.6** | **94.4** | 96.9 | 0.028 |
| MI | Deep Ens. | 85.1 | 89.0 | 92.4 | **95.4** | **0.034** |
| MI | SVE | **85.8** | **89.4** | **92.6** | 95.1 | 0.035 |

SVE and the Deep Ensemble perform almost identically. Under MSP they achieve the same AURC (0.028), and under MI they differ only marginally (0.035 vs 0.034). When SVE assigns high uncertainty to an in-distribution prediction, that prediction is misclassified at practically the same rate as for the Deep Ensemble, confirming that SVE does not simply produce overconfident predictions on ID data. As expected, MSP outperforms MI for selective classification, since ID misclassifications arise from both aleatoric and epistemic uncertainty and MSP reflects both, whereas MI isolates only the epistemic part. For OOD detection, where epistemic uncertainty is the more appropriate signal, MI becomes the more informative score.

# B. Ablation Study: Number of Ensemble Members

Both accuracy and calibration improve significantly from $M = 1$ to $M = 16$, on a complex task. See Tab.8. However, we observe that easier tasks like Flowers102 classification do not need many members; $M = 4$ is sufficient for optimal

accuracy and calibration.

*Table 8.* Effect of ensemble size $M$ on the ARC-Easy multiple-choice QA reasoning task using LLaMA-2-7B.

| $M$ | Acc(%)↑ | ECE(%)↓ | NLL↓ |
|---|---|---|---|
| 1 | 83.2 | 12.1 | 0.77 |
| 2 | 84.5 | 8.5 | 0.78 |
| 4 | 84.9 | 5.8 | 0.47 |
| 8 | 85.7 | 5.2 | 0.50 |
| 16 | **85.8** | **3.8** | **0.43** |

In practice, we recommend using $M = 4$ for standard fine-tuning tasks where computational cost is a concern, since it already provides sufficient accuracy and calibration for easier datasets. For harder reasoning or uncertainty-sensitive tasks, larger ensembles such as $M = 8$ or $M = 16$ can provide additional calibration gains, at the cost of linearly increased inference compute.

## C. Sensitivity Analysis: Singular-Value Initialization Scale

SVE relies on small member-specific perturbations of the pretrained singular values to break the symmetry between ensemble members. We ablate the perturbation scale $\sigma_{\text{init}}$ on Flowers102 with DINO ViT-S/16 and $M = 4$.

*Table 9.* Effect of $\sigma_{\text{init}}$ on Flowers102 with DINO ViT-S/16 and $M = 4$.

| $\sigma_{\text{init}}$ | Acc. (%)↑ | ECE (%)↓ | NLL↓ | MI↑ |
|---|---|---|---|---|
| 0 | 90.8 | 1.98 | 0.379 | 0.000 |
| 0.001 | 94.6 | 0.81 | 0.203 | 0.039 |
| 0.005 | **94.9** | 0.81 | **0.196** | 0.038 |
| 0.01 | 94.8 | **0.68** | 0.201 | 0.039 |
| 0.05 | 91.8 | 1.00 | 0.288 | 0.065 |
| 0.1 | 76.3 | 2.66 | 0.956 | **0.106** |

When $\sigma_{\text{init}} = 0$, all ensemble members are initialized identically and SVE collapses to a single singular-value fine-tuning model with zero mutual information. Small positive perturbations are sufficient to induce useful diversity and substantially improve both accuracy and calibration. Performance is stable for $\sigma_{\text{init}} \in [0.001, 0.01]$. Larger perturbations increase member disagreement but degrade accuracy and calibration, indicating that diversity is beneficial only when it remains close to the pretrained spectral structure.

## D. Ablation Study: Applying SVE to Subsets of Layers

The main experiments apply SVE to all selected linear transformations. To test whether all such layers are necessary, we evaluate partial-layer variants on Flowers102 with DINO ViT-S/16 and $M = 4$. Specifically, we compare applying SVE only to attention layers, only to MLP layers, and to both attention and MLP layers.

*Table 10.* Partial-layer SVE ablation on Flowers102 with DINO ViT-S/16 and $M = 4$.

| Scope | Trainable Parameter | Acc. (%)↑ | NLL↓ | MI↑ |
|---|---|---|---|---|
| Attention only | 110,592 | 94.94 | 0.1867 | 0.032 |
| MLP only | 129,024 | 94.84 | 0.1995 | **0.034** |
| Attention + MLP | 239,616 | **95.22** | **0.1840** | **0.034** |

Both attention-only and MLP-only SVE remain close to the full variant. Attention-only SVE already captures most of the ensemble diversity, reaching MI of 0.032 compared to 0.034 for the full attention+MLP setting, while using fewer than half the SVE parameters. Adding MLP layers provides a small additional improvement in accuracy and NLL. For generality and consistency, the main experiments use SVE for all selected linear transformations, but these results suggest that partial-layer SVE can be an effective lower-overhead variant.

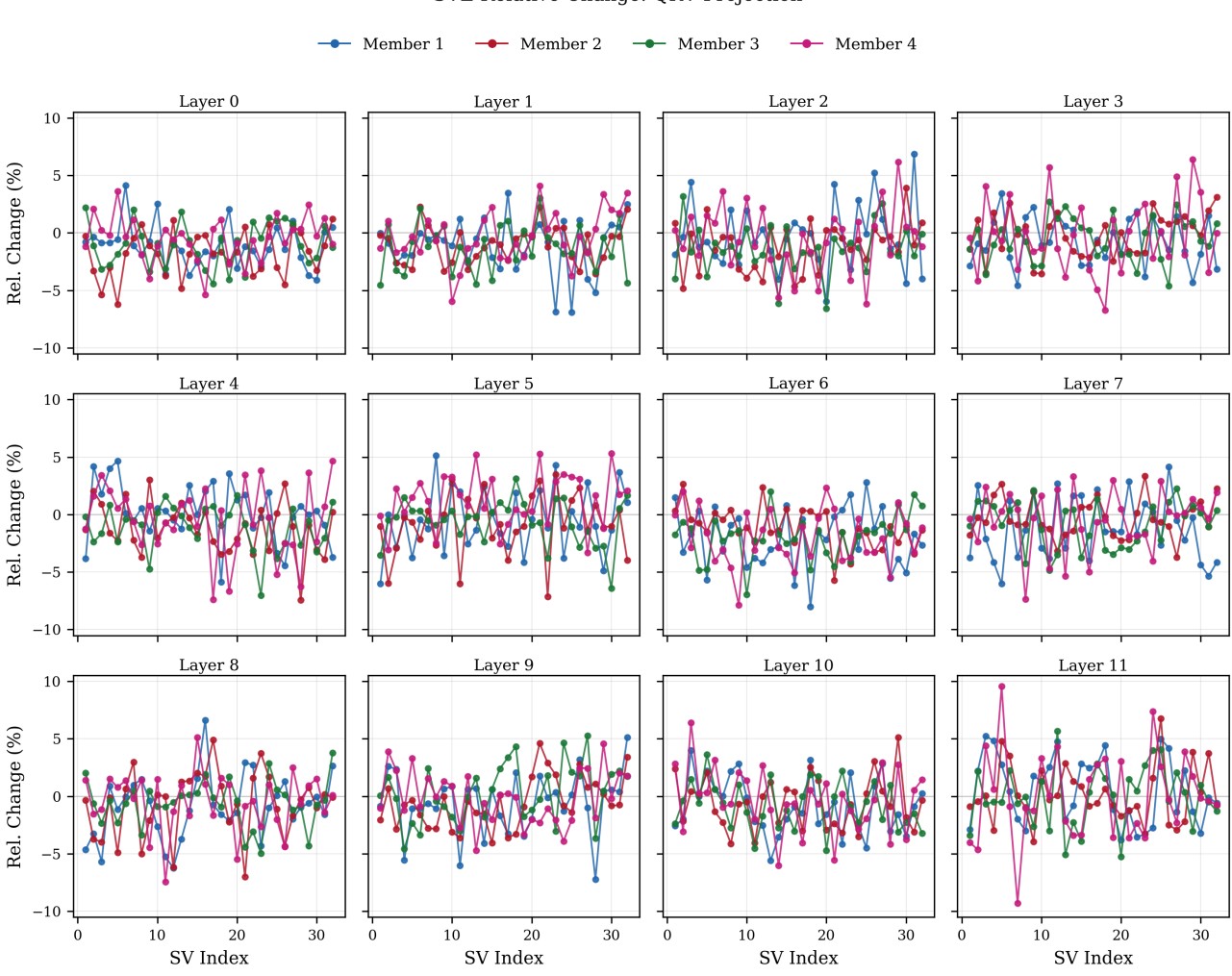

*Figure 4.* Top Singular Values' Relative Change (%) from Pretrained Weights for QKV attention projection layers of DINOv2 ViT-S/14 on Oxford Pets dataset.

## E. Singular Values Weight Diversity

We visualize the divergence of top singular values from their pretrained initialization across ensemble members in Figures 4, 5, and 6, showing Query-Key-Value projections, output projections, and fully-connected layers, respectively. The experiments use DINOv2-S/14, fine-tuned on Oxford Pets.

## F. Training Details

### F.1. Vision Experiments

For vision experiments, we use DINOv1 ViT-S/16 or DINOv2 ViT-S/14 as the frozen backbone and train only the ensemble-specific parameters and classification heads. Table 11 summarizes the hyperparameters.

### F.2. NLP Experiments

For ARC-Easy, we use LLaMA-2-7B (Touvron et al., 2023). For SST-2, we use BERT-base (Devlin et al., 2019). Table 11 summarizes the hyperparameters.

SVE Relative Change: Output Projection

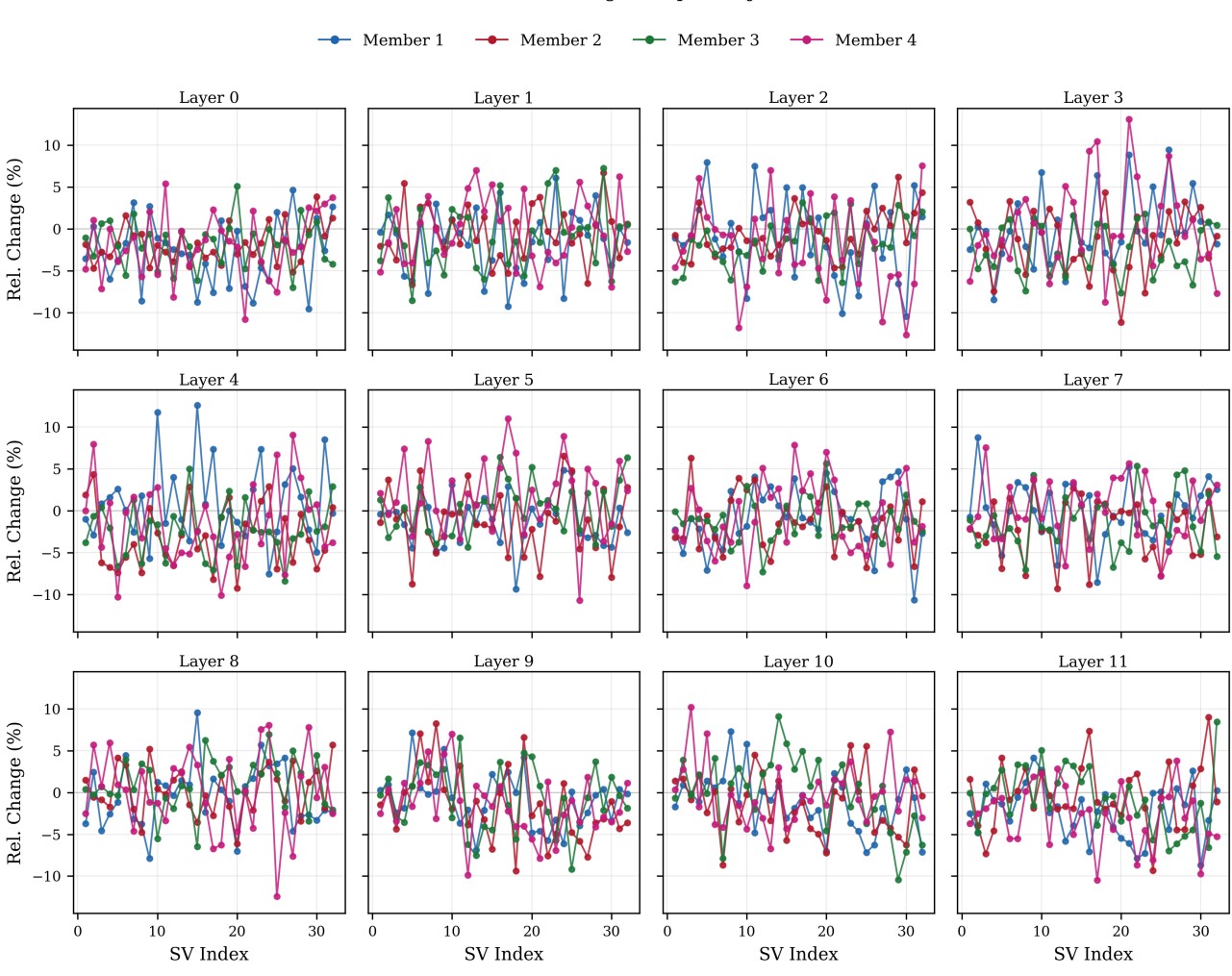

*Figure 5.* Top Singular Values' Relative Change (%) from Pretrained Weights for Output Projection layers of DINOv2 ViT-S/14 on Oxford Pets dataset.

### F.3. Baseline Configurations

For fair comparison, all methods use identical backbone architectures, optimizers, and training schedules. Some methods use a smaller learning rate, such as Single or Deep Ensemble, because they did converge with higher learning rates. We specified below if a different learning rate is used.

- **Deep Ensemble**: $M = 4$ independently trained models with different random seeds. Learning rate set to $1 \times 10^{-4}$.

- **MC Dropout**: Single model with dropout rate 0.05, using 10 forward passes at inference. Learning rate set to $1 \times 10^{-4}$.

- **LoRA-Ensemble**: Per-member LoRA adapters with rank $r = 8$ and $\alpha = 8$, applied to attention projections and fully connected layers. Learning rate set to $1 \times 10^{-3}$.

- **BatchEnsemble**: Rank-one perturbations with $M = 4$ members. Learning rate set to $1 \times 10^{-4}$.

All experiments are run with 3 random seeds and we report mean $\pm$ standard deviation.

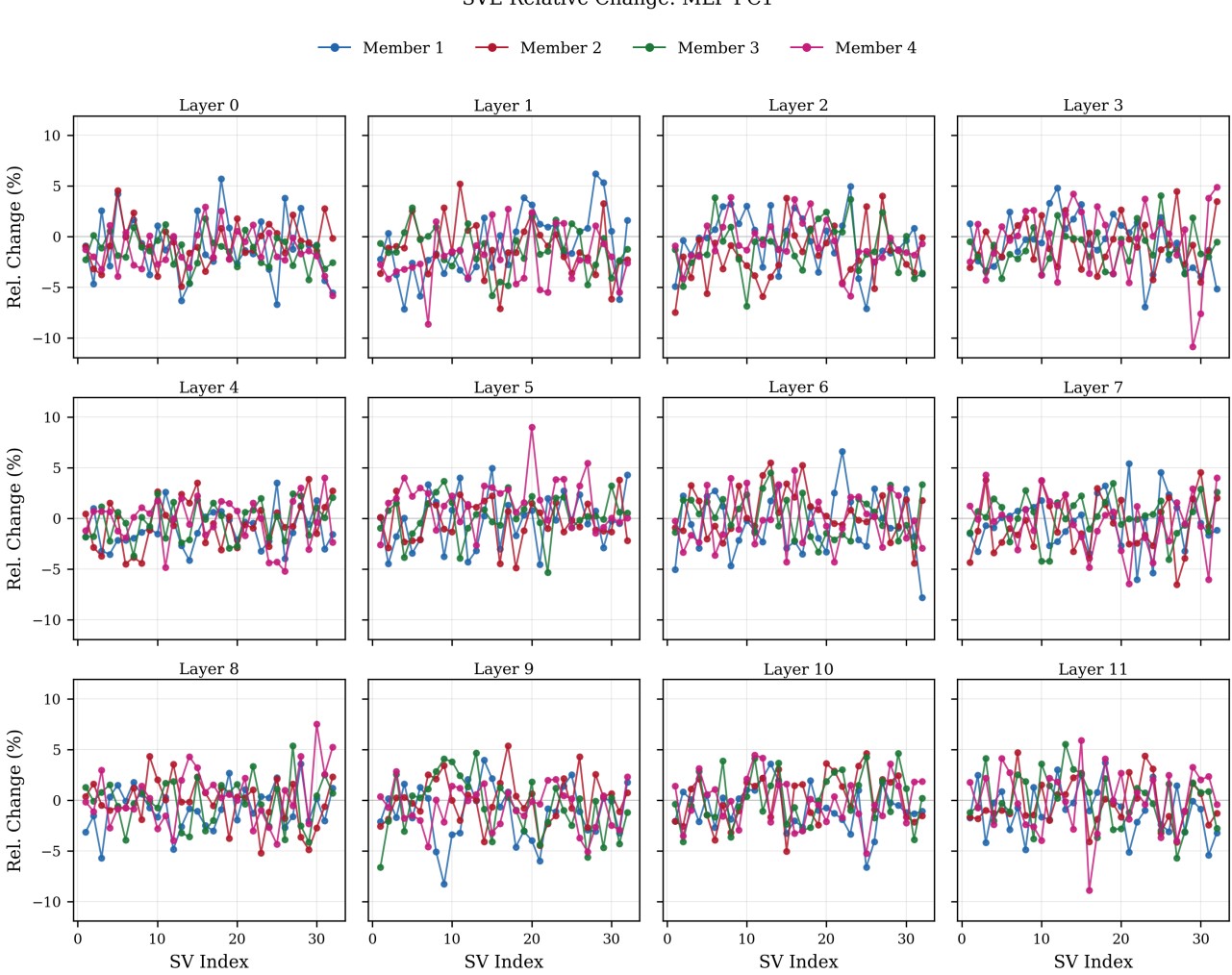

*Figure 6.* Top Singular Values' Relative Change (%) from Pretrained Weights for FC1 layers of DINOv2 ViT-S/14 on Oxford Pets dataset.

## G. Dataset Details

We evaluate SV-Ensemble on diverse downstream tasks spanning both natural language processing and computer vision, as outlined in the following. Table 12 further summarizes the key statistics of each dataset.

**ARC-Easy** (Clark et al., 2018) is a question-answering dataset, consisting of ∼ 5,200 grade school-level science questions from standardized tests. Each question has four or five answer choices. We use the Easy subset, which contains questions answerable with simple retrieval or reasoning. An example question is *Which change in the state of water particles causes the particles to become arranged in a fixed position?* with the following answer choices: 1. *boiling* 2. *melting* 3. *freezing* (correct) and 4. *evaporating*.

**SST-2.** The Stanford Sentiment Treebank (Socher et al., 2013) is a binary sentiment classification benchmark derived from movie reviews, commonly used to evaluate sentence-level sentiment understanding. We report results on the validation set (872 entries) following standard practice, as test labels are not publicly available.

**Flowers102.** The Oxford Flowers 102 dataset (Nilsback & Zisserman, 2008) is a fine-grained classification dataset containing images of 102 flower species commonly found in the United Kingdom. The dataset exhibits high intra-class variation and fine-grained distinctions between visually similar species. In total, the dataset comprises 8,189 images of

*Table 11.* Vision and NLP experiment hyperparameters.

| Hyperparameter | Flowers102 | CIFAR100 | DTD Texture | Oxford Pets | ARC-Easy | SST-2 |
|---|---|---|---|---|---|---|
| Backbone | DINO ViT-S/16 | DINOv ViT-S/16 DINOv2 ViT-S/14 | DINOv2 ViT-S/14 | DINOv2 ViT-S/14 | LLaMA-2-7B | BERT-base |
| Input resolution | 224×224 | 224×224 | 224×224 | 224×224 | - | - |
| Max sequence length | - | - | - | - | 512 | 128 |
| Optimizer | AdamW | AdamW | AdamW | AdamW | AdamW | AdamW |
| Learning rate | $1 \times 10^{-3}$ | $1 \times 10^{-4}$ | $1 \times 10^{-3}$ | $1 \times 10^{-3}$ | $3 \times 10^{-4}$ | $3 \times 10^{-4}$ |
| Weight decay | 0.05 | 0.05 | 0.05 | 0.05 | 0.01 | 0.01 |
| LR scheduler | Cosine | Cosine | Cosine | Cosine | Linear | Cosine |
| Gradient clipping | 1.0 | 1.0 | 1.0 | 1.0 | 1.0 | 1.0 |
| Warmup epochs | 5 | 5 | 5 | 0 | 3% | 5% |
| Batch size | 16 | 32 | 16 | 16 | 1 | 16 |
| Epochs | 10 | 10 | 10 | 200 iter. | 5 | 3 |
| Ensemble members ($M$) | 4 | 4 | 4 | 4 | 16 | 8 |
| SVE init std ($\sigma_{\text{init}}$) | 0.005 | 0.005 | 0.005 | 0.01 | 0.01 | 0.01 |

*Table 12.* Dataset statistics.

| Domain | Dataset | Classes | Train | Test |
|---|---|---|---|---|
| NLP | ARC-Easy | 4/5 choices | 2,251 | 2,376 |
| | SST-2 | 2 | 67,349 | 872 |
| Vision | Flowers102 | 102 | 1,020 | 6,149 |
| | CIFAR-100 | 100 | 50,000 | 10,000 |
| | DTD | 47 | 1,880 | 1,880 |
| | Oxford Pets | 37 | 3,680 | 3,669 |
| Robustness | CIFAR-10 (OOD) | 10 | – | 10,000 |
| | CIFAR-100-C | 100 | – | 10,000 |

which 6,149 are used for evaluation.

**CIFAR-100** (Krizhevsky, 2009) is a widely-used computer vision dataset containing 60,000 color images of resolution 32×32, across 100 fine-grained object classes grouped into 20 superclasses. The dataset is split into 50,000 training images and 10,000 test images. Due to the low native resolution, we resize images to match the input resolution required by the vision backbone.

**DTD.** The Describable Textures Dataset (Cimpoi et al., 2014) is a texture recognition benchmark consisting of 5,640 images annotated with 47 perceptual texture attributes such as banded, dotted, and woven. Each attribute is represented by 120 images collected from web sources, with image resolutions ranging from approximately 300×300 to 640×640. Images are annotated through crowd-sourcing and labeled with a primary (key) attribute as well as additional joint attributes. The dataset is provided with predefined train, validation, and test splits, each containing 40 images per class.

**Oxford Pets** (Parkhi et al., 2012) is an image classification dataset containing images of 37 pet breeds, including 25 dog breeds and 12 cat breeds, with roughly 200 images per class. The images exhibit large variations in scale, pose, and illumination.

**CIFAR-10 / CIFAR-100-C.** For out-of-distribution detection, we use CIFAR-10 (Krizhevsky, 2009) as a semantically shifted dataset relative to models trained on CIFAR-100. For robustness under distribution shift, we use CIFAR-100-C (Hendrycks & Dietterich, 2019), which applies 19 corruption types (e.g., Gaussian noise, blur, weather effects) at five severity levels to the CIFAR-100 test set.

# H. Definitions of Evaluation Metrics

We evaluate our models using a range of standard metrics commonly employed in probabilistic deep learning and related evaluation settings. This section summarizes the precise formulations used in our implementations.

## H.1. Accuracy

Accuracy measures the proportion of correctly classified samples and is computed as

$$\text{Acc} = \frac{1}{N} \sum_{i=1}^{N} \mathbf{1}(\hat{y}_i = y_i), \tag{8}$$

where $y_i$ denotes the ground-truth label of sample $i$, $\hat{y}_i$ is the corresponding predicted label, $N$ is the total number of samples, and $\mathbf{1}(\cdot)$ is the indicator function.

## H.2. Expected Calibration Error (ECE)

The Expected Calibration Error (ECE) is a standard metric for evaluating the calibration quality of neural network predictions. We follow the definition proposed by (Guo et al., 2017). ECE quantifies the expected discrepancy between predictive confidence and empirical accuracy across a set of confidence bins. In our experiments, we partition the interval $[0, 1]$ into $M = 15$ equally spaced bins.

For each bin $B_m$, the accuracy and confidence are defined as

$$\text{Acc}(B_m) = \frac{1}{|B_m|} \sum_{i \in B_m} \mathbf{1}(\hat{y}_i = y_i), \tag{9}$$

$$\text{Conf}(B_m) = \frac{1}{|B_m|} \sum_{i \in B_m} \hat{p}_i, \tag{10}$$

where $\hat{p}_i$ denotes the predicted confidence of sample $i$. The Expected Calibration Error is then given by

$$\text{ECE} = \sum_{m=1}^{M} \frac{|B_m|}{N} |\text{Acc}(B_m) - \text{Conf}(B_m)|. \tag{11}$$

## H.3. Negative Log-Likelihood (NLL)

The Negative Log-Likelihood (NLL) evaluates the quality of probabilistic predictions and is defined as

$$\text{NLL} = -\frac{1}{N} \sum_{i=1}^{N} \sum_{j=1}^{C} y_{i,j} \log \hat{p}_{i,j} = -\frac{1}{N} \sum_{i=1}^{N} \log \hat{p}_i, \tag{12}$$

where $N$ is the number of samples, $C$ is the number of classes, $y_{i,j}$ equals 1 if the true label of sample $i$ is class $j$, and 0 otherwise; $\hat{p}_{i,j}$ is the predicted probability assigned to class $j$ for sample $i$.

## H.4. Brier Score

We compute the Brier score following the definition introduced by (Brier, 1950). The metric is given by

$$\text{BS} = \frac{1}{N} \sum_{i=1}^{N} \sum_{j=1}^{C} (\hat{p}_{i,j} - y_{i,j})^2, \tag{13}$$

where $N$ is the number of samples, $C$ the number of classes, $y_{i,j}$ indicates whether class $j$ is the true label of sample $i$, and $\hat{p}_{i,j}$ is the corresponding prediction of its class probability.

## H.5. Area Under the Receiver Operating Characteristic Curve (AUROC)

The Area Under the Receiver Operating Characteristic Curve (AUROC) measures the ability of a binary classifier to discriminate between positive and negative samples (Hanley & McNeil, 1982). In our out-of-distribution (OOD) detection experiments, in-distribution samples are treated as positives, while out-of-distribution samples form the negative class.

The ROC curve plots the true positive rate (TPR) against the false positive rate (FPR) across varying decision thresholds. These quantities are defined as

$$\text{TPR} = \frac{\text{TP}}{\text{TP} + \text{FN}}, \tag{14}$$

$$\text{FPR} = \frac{\text{FP}}{\text{FP} + \text{TN}}, \tag{15}$$

where TP, FP, FN, and TN denote true positives, false positives, false negatives, and true negatives, respectively. The AUROC is then computed as

$$\text{AUROC} = \int_0^1 \text{TPR}(\text{FPR}) \, d\text{FPR}. \tag{16}$$

An AUROC of 1 corresponds to perfect discrimination, while a value of 0.5 indicates performance equivalent to random guessing.

### H.6. Area Under the Precision–Recall Curve (AUPRC)

The Area Under the Precision–Recall Curve (AUPRC) evaluates binary classification performance with a particular focus on the positive class (Davis & Goodrich, 2006). As in the AUROC setting, in-distribution samples are treated as positives in our OOD experiments.

The precision–recall curve is obtained by plotting precision against recall at different thresholds, where

$$\text{Precision} = \frac{\text{TP}}{\text{TP} + \text{FP}}, \tag{17}$$

$$\text{Recall} = \frac{\text{TP}}{\text{TP} + \text{FN}}. \tag{18}$$

The AUPRC is defined as the integral of precision with respect to recall:

$$\text{AUPRC} = \int_0^1 \text{Precision}(\text{Recall}) \, d\text{Recall}. \tag{19}$$

Higher AUPRC values indicate improved performance, particularly on imbalanced datasets.

### H.7. False Positive Rate at 95% True Positive Rate (FPR@95%TPR)

We additionally report the false positive rate at a fixed true positive rate of 95% (FPR@95%TPR). This metric captures the proportion of negative samples incorrectly classified as positive when maintaining a high detection rate for positive samples. Lower values indicate better performance.

Formally, let $\tau$ be a threshold such that

$$\text{TPR}(\tau) = \frac{\text{TP}(\tau)}{\text{TP}(\tau) + \text{FN}(\tau)} = 0.95. \tag{20}$$

The FPR@95%TPR is then defined as

$$\text{FPR@95\%TPR} = \frac{\text{FP}(\tau)}{\text{FP}(\tau) + \text{TN}(\tau)}. \tag{21}$$

### H.8. Area Under the Risk–Coverage Curve (AURC)

The Area Under the Risk–Coverage curve (AURC) summarises the quality of an uncertainty score for selective classification. Samples are ranked by uncertainty, and for each coverage level $c \in (0, 1]$, the $\lfloor cN \rfloor$ most certain samples are retained while the remainder is rejected. The risk at coverage $c$ is the empirical error rate on the retained set $\mathcal{R}_c$:

$$\text{Risk}(c) = \frac{1}{|\mathcal{R}_c|} \sum_{i \in \mathcal{R}_c} \mathbf{1}(\hat{y}_i \neq y_i). \tag{22}$$

The AURC is then the integral of risk over coverage:

$$\text{AURC} = \int_0^1 \text{Risk}(c) \, dc. \tag{23}$$

Lower AURC indicates a more informative uncertainty score, since the most uncertain predictions are then more likely to be the misclassified ones.

