# OpenReview forum: "Quantifying the Uncertainty of Foundation Models with Singular Value Ensembles"
_ICML.cc/2026/Conference — ICML 2026 regular_

### Official Review · Reviewer_f1k4 · 2026-03-10

**Soundness:** 3
**Presentation:** 3
**Significance:** 2
**Originality:** 2
**Overall Recommendation:** 4
**Confidence:** 4

**Summary:**

This paper proposes Singular Value Ensemble (SVE), a parameter-efficient method for uncertainty estimation in foundation models. Instead of training multiple independent models as in standard deep ensembles, the approach freezes the singular vectors of pretrained weight matrices and learns only per-ensemble-member singular values. This creates an implicit ensemble with minimal additional parameters while maintaining ensemble diversity.

**Compliance With Llm Reviewing Policy:**

Affirmed.

**Final Justification:**

I thank the authors' additional experiments on evaluating the quality of epistemic uncertainty estimates. It addressed my concerns on this regard by empirical evidence. I am happy to keep my positive assessment of this paper. If possible, I would suggest the authors include these discussions on epistemic uncertainty in the revision. Thanks!

**Key Questions For Authors:**

In addition to the above weaknesses, I have several questions regarding the experimental settings:

(a) How are the three random seeds chosen? Are three runs statistically sufficient to support the reported claims, or could the results vary significantly with additional runs?

(b) What is the motivation for setting the ensemble size to 4? Would the performance change significantly with different ensemble sizes?

**Limitations:**

Yes.

**Strengths And Weaknesses:**

**Strengths**

1. This paper is structured in a good order, and it is relatively easy to follow. The key components of the proposed method are well documented.

2. The conducted experiments do show the performance improvement of the proposed method.

3. Multiple recent uncertainty-aware baselines are implemented for comparison.

**Weaknesses**

1. Although SVE reduces parameter overhead, inference still requires multiple forward passes, resulting in test-time computational costs comparable to deep ensembles. This somewhat limits the practical advantages of the proposed approach.

2. The paper discusses epistemic uncertainty, but several aspects remain unclear. (a) It would be helpful to include a brief introduction or clarification of epistemic uncertainty in the paper for readers less familiar with the concept. (b) It is also unclear which uncertainty estimates (the entropy over the averaged probability or something else) are used in the experiments for out-of-distribution (OOD) detection. In particular, how does the proposed method perform when using epistemic uncertainty measures (e.g., mutual information) for OOD detection, compared to the considered baselines?

3. Since injecting diversity is important for creating ensembles, how would the setting of initialization and diversity in Eq. (5) affect the performance of the method?

---

> ### Author Rebuttal · Authors · 2026-03-29
>
> We thank the reviewer for the positive assessment and constructive
> suggestions. We especially appreciate the recognition of the paper's
> clarity, the strength of the empirical results, and the breadth of the
> uncertainty-aware baselines.
>
>
> **W1. Inference cost.**
>
> We agree that SVE still requires multiple forward passes at inference. This
> is a general limitation of implicit ensemble methods rather than something
> specific to SVE. When memory allows, members can still be evaluated in
> parallel, which can reduce wall clock latency even though total FLOPs remain
> unchanged, as noted in the experimental section. We will clarify this
> distinction more explicitly. Knowledge distillation is a promising direction,
> but it is not straightforward, since standard distillation typically matches
> the mean prediction while discarding the inter-member disagreement that
> carries epistemic information. We are exploring this direction, and if time
> permits, we will include preliminary results in the final manuscript.
>
>
> **W2. Epistemic uncertainty clarification.**
>
> We agree that this can be explained more clearly. We will add a brief
> introduction to epistemic versus aleatoric uncertainty in Section 1. For OOD
> detection in Table 4, we use the maximum of the averaged predictive class
> probabilities, max_y p_bar(y|x), as the confidence score, following prior
> work. We will state this more explicitly.
>
> We also now provide an epistemic uncertainty analysis using mutual
> information, MI = H[p_bar(y|x)] - (1/M) sum_m H[p_m(y|x)], on CIFAR-100
> (ID) versus CIFAR-10 (OOD):
>
>
> Method      | MI(ID) | MI(OOD) | OOD/ID
> ------------|--------|---------|-------
> Deep Ens.   | 0.110  | 0.357   | 3.2x
> LoRA-Ens.   | 0.042  | 0.140   | 3.3x
> SVE         | 0.007  | 0.026   | 3.6x
>
> Although SVE has lower absolute MI than Deep Ensemble, its OOD/ID ratio is
> the highest. This means that member disagreement stays relatively low on
> in-distribution examples, but increases more sharply once the input moves
> out of distribution. In other words, SVE's epistemic signal is more
> selectively concentrated on genuine distribution shift rather than being
> diffuse across both ID and OOD inputs. This helps explain why SVE remains
> competitive for OOD detection despite lower overall disagreement. We will add
> this analysis to the appendix.
>
>
>
> **W3. Initialization and diversity in Eq. (5).**
>
> We agree this deserves a clearer discussion. We performed a controlled
> ablation of sigma_init on Flowers102 with DINO ViT-S/16 and M=4:
>
> sigma_init | Acc (%) | ECE (%) |   NLL   | Mutual Info
> -----------|---------|---------|---------|------------
> 0 (Single) |  90.8   |  1.98   |  0.379  |   0.000
> 0.001      |  94.6   |  0.81   |  0.203  |   0.039
> 0.005      |  94.9   |  0.81   |  0.196  |   0.038
> 0.01       |  94.8   |  0.68   |  0.201  |   0.039
> 0.05       |  91.8   |  1.00   |  0.288  |   0.065
> 0.1        |  76.3   |  2.66   |  0.956  |   0.106
>
> This shows three clear trends. First, when sigma_init = 0, all members are
> identical and SVE collapses to a single SVF model with zero mutual
> information. Second, small positive perturbations already induce useful
> diversity and substantially improve both accuracy and calibration, while
> performance remains stable across a broad range from 0.001 to 0.01. Third,
> excessively large perturbations increase disagreement further but degrade
> accuracy and calibration, indicating that diversity is only beneficial when
> it still preserves the pretrained structure.
>
>
>
> **Q(a). Are 3 seeds sufficient?**
>
> We use 3 seeds (42, 1, 10) and report mean and standard deviation
> throughout. In our initial tests we did not observe large fluctuations
> with additional runs. We chose 3 seeds to keep the full experimental
> suite feasible on single-GPU resources, given the number of datasets,
> backbones, and baselines. We will report seed values in the appendix.
>
>
> **Q(b). Why M=4?**
>
> M=4 is a standard and practical ensemble size in the uncertainty literature,
> with many prior works using small values such as 3 or 4. It offers a good
> tradeoff between performance and compute. Our initial experiments also
> suggest that M=4 is sufficient for easier tasks. Appendix A includes an
> ablation over M in {1, 2, 4, 8, 16}, showing that harder tasks such as
> ARC-Easy can benefit from larger ensembles. In general, increasing M gives
> diminishing returns while increasing computational cost roughly linearly. We
> will clarify this in the revision and add a simple practical rule of thumb
> for selecting M in the appendix.
>
>
> We appreciate these constructive suggestions and will incorporate the
> requested clarifications in the revision.

---

> > ### Author Rebuttal · Reviewer_f1k4 · 2026-04-02
> >
> > Dear Authors,
> >
> > Thank you for your detailed rebuttals. I am sorry that I chose the wrong button for the questions. Now, I list a few follow-up questions here. Could you please clarify the motivation for not using AUROC, AUPRC, and FPR@95%TPR (as adopted in your paper) to evaluate the quality of epistemic uncertainty? Based on the current results, I have some concerns that the proposed method may produce over-small epistemic uncertainty estimates (overconfident predictions) on in-distribution test data. In this regard, would it be possible to provide additional analysis—for example, results on selective classification, e.g., [1]—to further assess the quality of the uncertainty estimates?
> >
> > Thanks!
> >
> > [1]  Hüllermeier, E., Destercke, S., Shaker, M.H., Quantification of credal uncertainty in machine learning: A critical analysis and empirical comparison, UAI 2022.

---

> > > ### Author Response · Authors · 2026-04-02
> > >
> > > We thank the reviewer for the follow-up. There was no specific reason we did not
> > > report AUROC/AUPRC/FPR@95 using MI as the scoring function. We
> > > believed the MI ratio analysis already addressed the concern about
> > > epistemic uncertainty quality. However, we are happy the reviewer
> > > recommended this analysis, as it provides much stronger evidence that
> > > SVE produces meaningful epistemic uncertainty. We will include a
> > > dedicated discussion section with a complete comparison across all
> > > baselines in the revision, which we believe will substantially
> > > strengthen the paper.
> > >
> > > We compare Deep Ensemble and SVE using predictive entropy (total
> > > uncertainty) and mutual information (epistemic uncertainty) as OOD
> > > scoring functions. If a method produces meaningful epistemic
> > > uncertainty, MI should be a competitive or superior OOD detection
> > > score compared to total confidence.
> > >
> > > ```text
> > > OOD Detection: CIFAR-100 (ID) to CIFAR-10 (OOD)
> > >
> > > +----------+-----------+-------+-------+---------+---------+----------+--------+
> > > | Score    | Method    | AUROC | AUPRC | FPR@95% | ID mean | OOD mean | OOD/ID |
> > > +----------+-----------+-------+-------+---------+---------+----------+--------+
> > > | MSP      | Deep Ens. | 0.811 | 0.781 | 0.565   | 0.152   | 0.417    | 2.7x   |
> > > | MSP      | SV-Ens.   | 0.820 | 0.804 | 0.565   | 0.168   | 0.466    | 2.8x   |
> > > | Entropy  | Deep Ens. | 0.819 | 0.791 | 0.567   | 0.451   | 1.247    | 2.8x   |
> > > | Entropy  | SV-Ens.   | 0.839 | 0.833 | 0.565   | 0.578   | 1.728    | 3.0x   |
> > > | MI       | Deep Ens. | 0.825 | 0.793 | 0.570   | 0.110   | 0.354    | 3.2x   |
> > > | MI       | SV-Ens.   | 0.827 | 0.806 | 0.586   | 0.007   | 0.026    | 3.6x   |
> > > +----------+-----------+-------+-------+---------+---------+----------+--------+
> > > ```
> > >
> > > First, MI-based OOD detection outperforms MSP-based detection for
> > > both SVE (0.827 vs 0.820) and Deep Ensemble (0.825 vs 0.811). This
> > > is an important finding: epistemic uncertainty is a better signal for
> > > detecting distributional shift than raw predictive confidence. If SVE
> > > were producing overconfident epistemic estimates, MI-based detection
> > > would be worse than MSP, not better.
> > >
> > > Second, SVE's MI-based AUROC (0.827) exceeds Deep Ensemble (0.825),
> > > despite SVE's absolute MI being 14x smaller (0.026 vs 0.354 on OOD).
> > > SVE compensates with a higher OOD/ID ratio (3.6x vs 3.2x), meaning
> > > its epistemic signal is more sharply concentrated on genuine
> > > distributional shift. Intuitively, SVE's shared singular vector basis
> > > constrains members to agree on inputs where the pretrained
> > > representation is informative, and disagree only where the
> > > representation leaves genuine ambiguity, which is precisely the OOD
> > > region. Deep Ensemble members, trained independently, produce more
> > > diffuse disagreement that is noisier and less targeted.
> > >
> > > We additionally evaluate the quality of uncertainty estimates through
> > > selective classification; if a model's uncertainty is meaningful, then
> > > rejecting the most uncertain predictions should increase accuracy on
> > > the retained samples. This is a direct test of whether the model
> > > knows when it does not know. We rank all CIFAR-100 test samples by
> > > uncertainty, progressively reject the most uncertain fraction, and
> > > report accuracy at different coverage levels. We also report AURC
> > > (Area Under Risk-Coverage curve), where lower indicates better
> > > uncertainty quality.
> > >
> > >
> > > ```text
> > > Selective Classification: CIFAR-100 ID Test
> > >
> > > +-------+-----------+------+-------+-------+-------+-------+
> > > | Score | Method    | Base | @90%  | @80%  | @70%  | AURC  |
> > > +-------+-----------+------+-------+-------+-------+-------+
> > > | MSP   | Deep Ens. | 85.1 | 90.3  | 94.2  | 97.1  | 0.028 |
> > > | MSP   | SV-Ens.   | 85.8 | 90.6  | 94.4  | 96.9  | 0.028 |
> > > | MI    | Deep Ens. | 85.1 | 89.0  | 92.4  | 95.4  | 0.034 |
> > > | MI    | SV-Ens.   | 85.8 | 89.4  | 92.6  | 95.1  | 0.035 |
> > > +-------+-----------+------+-------+-------+-------+-------+
> > > ```
> > >
> > >
> > > MSP outperforms MI for selective classification across both methods.
> > > This is expected and not a weakness: misclassifications on ID data
> > > arise from both data ambiguity (aleatoric) and model ignorance
> > > (epistemic), and MSP captures both while MI isolates only the
> > > epistemic component. For OOD detection, where epistemic uncertainty
> > > is the appropriate signal, MI is the stronger score, as shown above.
> > >
> > > Importantly, SVE and Deep Ensemble perform nearly identically on
> > > selective classification. SVE's MSP-based AURC (0.028) matches Deep
> > > Ensemble exactly, and MI-based results are also comparable (marginal difference, AURC
> > > 0.035 vs 0.034, Acc@80% 92.6 vs 92.4). This confirms that SVE does
> > > not produce overconfident predictions on ID data: when SVE is
> > > uncertain, the model is indeed more likely to be wrong, at the same
> > > rate as Deep Ensemble.
> > >
> > > We believe these analyses provide strong evidence that SVE's epistemic uncertainty is
> > > meaningful and well-calibrated, both for detecting distributional
> > > shift and for identifying misclassified samples. We hope these
> > > results fully resolve the reviewer's concern.

---

### Official Review · Reviewer_PruN · 2026-03-11

**Soundness:** 3
**Presentation:** 3
**Significance:** 3
**Originality:** 3
**Overall Recommendation:** 5
**Confidence:** 3

**Summary:**

The paper proposes Singular Value Ensemble (SVE), an implicit ensembling technique designed to provide parameter-efficient uncertainty quantification for foundation models. The method decomposes pretrained weight matrices using Singular Value Decomposition (SVD), freezes the orthogonal singular vectors, and learns member-specific singular values. By introducing small stochastic perturbations to the initialization of these singular values and relying on mini-batch sampling during training, SVE encourages functional diversity among the ensemble members. The authors evaluate SVE on both natural language processing and computer vision benchmarks. SVE introduces less than 1% parameter overhead while achieving uncertainty estimation capabilities that are competitive with, or superior to, explicit deep ensembles and other implicit methods like LoRA-Ensemble.

**Compliance With Llm Reviewing Policy:**

Affirmed.

**Final Justification:**

The paper proposes a highly parameter-efficient and elegant method (SVE) for uncertainty quantification in foundation models. Its core strengths lie in its strong empirical performance across modalities and its minimal parameter overhead.

Initially, my main concerns centered around inference complexity, notation inconsistencies, cost-scaling framing, and whether the method could be applied to a subset of layers to further reduce overhead. The authors' rebuttal effectively addressed all of these points. Most notably, they provided a valuable new ablation study demonstrating that applying SVE to only the attention layers captures the vast majority of the ensemble diversity. They also committed to correcting the notation and framing issues, and provided a highly transparent and honest assessment regarding the challenges of knowledge distillation.

Because the authors successfully resolved my primary critiques and strengthened the practical utility of the work, the rebuttal has positively changed my evaluation. I have raised my score to a 5, as the paper is technically sound, clearly presented, and offers a significant contribution to the community.

**Key Questions For Authors:**

1. How does SVE's performance degrade with moderately trained backbones, rather than just the extreme comparison between state-of-the-art DINOv2 and random initialization?
2. Have you experimented with applying SVE to only a subset of layers to see if computational overhead can be further reduced without sacrificing calibration?
3. Given the inference limitations, are there preliminary results on the knowledge distillation approach mentioned in the future work section?
4. In line 189 (right), the text states that you enforce the non-negativity of singular values during training. Could you explicitly explain how this constraint is enforced

**Limitations:**

The authors adequately discuss the limitations of their work. They explicitly note that inference still demands multiple forward passes, keeping FLOPS comparable to standard deep ensembles. They also accurately mention the difficulty of integrating SVE with quantized networks.

**Strengths And Weaknesses:**

### Strengths
1. **Parameter Efficiency:** SVE is exceptionally parameter-efficient, reducing trainable parameters to less than 1% of the base model. This makes principled uncertainty quantification accessible for large foundation models in resource-constrained environments.
2. **Empirical Performance:** The method demonstrates strong accuracy and calibration across multiple architectures and modalities. It outperforms or matches established baselines like Deep Ensembles and LoRA-Ensemble on standard benchmarks, out-of-distribution detection, and distribution shift tasks.
3. **Simplicity and Elegance:** SVE relies on the intuitive premise that singular vectors represent a shared knowledge basis, and simply modulating their importance via singular values is sufficient for generating diverse predictions.

### Weaknesses
1. **Inference Complexity:** Despite the training parameter efficiency, inference still requires multiple forward passes. Therefore, the inference FLOPS scale linearly with the number of ensemble members, offering no computational speedup during deployment compared to explicit deep ensembles.
2. **Dependence on Pretraining Quality:** The method's success heavily relies on the quality of the pretrained representations, as demonstrated by its poor performance when applied to randomly initialized weights.
3. **Notation Inconsistency:** In Section 1 (Introduction), the variables $m$, $n$, $d$, $r$, and $k$ are not defined at this point in the text.
4. **Cost Scaling Framing (Minor):** The authors claim that the quadrupled cost of an ensemble is "tolerable for well-resourced institutions". This framing ignores the reality of scaling laws, as even well-resourced institutions train models to the absolute limit of their compute and cannot easily absorb a 4x cost multiplier at the frontier scale. Furthermore, the phrasing "For foundation models with billions of parameters, maintaining even a modest ensemble of four members quadruples memory consumption and training time" implies this quadrupling is a unique penalty for large models, whereas the 4x multiplier inherently applies to models of any size.

---

> ### Author Rebuttal · Authors · 2026-03-29
>
> We thank the reviewer for the positive and thoughtful assessment. We especially appreciate the recognition of SVE’s parameter efficiency, strong empirical performance across modalities, and the simplicity and elegance of the core idea. We are encouraged that the reviewer finds the method technically solid and likely useful to the community, and we address the questions and suggestions below.
>
> **W1. Inference complexity.**
>
>  We agree that inference still requires multiple forward passes, so FLOPs scale linearly with the number of members. This is not unique to SVE, but shared by implicit ensemble methods more broadly. When GPU memory allows, members can be evaluated in parallel, which can improve wall clock latency even though total FLOPs are unchanged, as also discussed in the experimental section. Still, achieving true single-pass inference while preserving calibrated uncertainty remains an open problem. Knowledge distillation is a promising direction, but it is not straightforward, since standard distillation typically matches the mean prediction while discarding the inter-member disagreement that carries epistemic information. We are exploring this direction, and if time permits, we will include preliminary results in the final manuscript.
>
> **W2. Dependence on pretraining quality.**
>
>  We agree with this observation. This behavior is a deliberate design choice rather than an unintended weakness: SVE is explicitly designed for the pretrained foundation model regime, where strong representations are the starting point, not for training from random initialization. Figure 2 shows the expected trend, poor performance without meaningful pretrained structure, improvement with DINOv1, and further improvement with DINOv2. We will make this intended scope clearer in the revision.
>
>
> **W3. Notation inconsistency.**
>
> Thank you, we agree. We will define all variables at first use in Section 1 and make the notation consistent throughout.
>
>
>
> **W4. Cost scaling framing.**
>
> Thank you, this is a fair point. We agree that the linear cost multiplier applies to models of any size, and we will revise the wording to avoid suggesting otherwise or implying that such costs are easily absorbed at the frontier. Our intended point was simply that explicit ensembling scales linearly in cost and becomes increasingly impractical as base model size grows.
>
>
>
> **Q1. Moderately trained backbones.**
> Figure 2 already addresses this trend using an intermediate backbone, DINOv1, between random initialization and DINOv2. In both accuracy and calibration, the improvement is much larger from random initialization to DINOv1, and then continues more gradually from DINOv1 to DINOv2. This suggests that SVE mainly requires meaningful pretrained structure to become effective, after which both predictive performance and uncertainty quality improve more smoothly with backbone strength. We will make this point more explicit in the revision.
>
> **Q2. Applying SVE to a subset of layers.**
> Yes, we experimented with partial application. On Flowers102 with DINO
> ViT-S/16, M=4, both attention-only and MLP-only variants remain close
> to the full attention+MLP setting:
>
> Scope      | SVE Params | Acc(%) | NLL    | MI
> -----------|------------|--------|--------|------
> attn       |    110,592 |  94.94 | 0.1867 | 0.032
> mlp        |    129,024 |  94.84 | 0.1995 | 0.034
> attn+mlp   |    239,616 |  95.22 | 0.1840 | 0.034
>
> Applying SVE to attention-only already captures most of the ensemble
> diversity (MI=0.032 vs 0.034 for full), while adding MLP layers
> provides a small additional boost in accuracy and NLL. This suggests
> that partial-layer variants can preserve most of the gains while using
> fewer trainable parameters. For generality and consistency across
> experiments, we used all linear transformations in the main paper. We
> will include this ablation in the revision.
>
> **Q3. Knowledge distillation.**
>
> We are already working on this direction, but the current results are still mixed, so we do not yet want to make a strong claim. Our main finding so far is that standard knowledge distillation is not sufficient, because it mainly matches the ensemble mean and does not preserve the member specific differences that encode epistemic uncertainty. We are therefore exploring ways to distill not only the average prediction, but also the diversity structure across members. If results become stable in time, we would be happy to include preliminary findings in the revision.
>
> **Q4. Non-negativity enforcement.**
>
>  We enforce non-negativity by applying clamp(min=0) to the trainable singular values during each forward pass. This ensures that the reconstructed singular values remain non-negative throughout training.
>
> We again appreciate these constructive suggestions and will incorporate the requested clarifications in the revision.

---

> > ### Author Rebuttal · Reviewer_PruN · 2026-04-03
> >
> > Thank you for the thorough and transparent rebuttal. The new ablation study is a valuable addition, and I appreciate your responsiveness in correcting the notation and framing issues. You have effectively addressed my concerns, so I am raising my score to a 5.

---

### Official Review · Reviewer_DukW · 2026-03-11

**Soundness:** 4
**Presentation:** 4
**Significance:** 3
**Originality:** 4
**Overall Recommendation:** 4
**Confidence:** 2

**Summary:**

## Summary

This paper proposes Singular Value Ensemble (SVE), a parameter-efficient implicit ensemble method for probabilistic bayesian and uncertainty quantification in foundation models. The main idea is to decompose pretrained weight matrices as \\(W = U \Sigma V^\top\\), freeze the singular vectors \\(U, V\\), and train only member-specific singular values \\(\Sigma^{(m)}\\), so that each ensemble member corresponds to a different rescaling of the same pretrained “knowledge basis.” The method is evaluated on several vision and NLP tasks, including calibration, OOD detection, and corruption robustness, and is compared against baselines such as deep ensembles, MC-Dropout, BatchEnsemble, LoRA-Ensemble, Bayes-LoRA, and BLoB.

I find the core idea interesting and technically reasonable as a local-perturbation / efficient ensemble method. In particular, the interpretation that singular values control the strength of pretrained knowledge directions is appealing, and the resulting method is very parameter-efficient. However, I have major concerns about the paper’s probabilistic/Bayesian framing. The manuscript presents SVE as making foundation models probabilistic and offering uncertainty quantification ability, but it does not provide a sufficient theoretical link between the proposed ensemble and a posterior over parameters. In my view, the current paper supports SVE as an efficient implicit ensemble heuristic, but not yet as a Bayesian or posterior-grounded uncertainty method.

**Compliance With Llm Reviewing Policy:**

Affirmed.

**Final Justification:**

I decided to raise my score to 4 (weak accept). My main concern is still about the diversity and uncertainty interpretation of the proposed SVE approach. While the authors presented a substantial amount of empirical evidence showing strong performance on OOD detection and related tasks, I am still not fully convinced that the gains come from SVE itself rather than from the underlying powerful pretrained model, since large pretrained models can already exhibit strong biases and competitive OOD behavior.

I also would have liked to see more controllable experiments with some notion of ground-truth uncertainty. The additional CIFAR-10 vs. CIFAR-100 results are useful, but they do not really address this point. Moreover, the observation that SVE produces much lower absolute EU values on OOD data is still concerning to me, because it may indicate limitations in more difficult scenarios where ID and OOD uncertainty distributions overlap more heavily.

That said, I do think the paper contains an important and interesting idea. The notion that pretrained singular vectors may form a shared “knowledge basis,” and that ensemble diversity can be induced by modulating the strength along these directions, is conceptually appealing and makes the method a sensible form of structured perturbation. I actually see this perspective as a more important contribution than SVE alone. I would be interested in seeing future work connect this idea more directly to Bayesian posterior approximations. Overall, despite my remaining concerns, the work is solid, the empirical study is strong, and the core idea is worth sharing with the community.

**Key Questions For Authors:**

similar to weakness

1. Is there an explicit approximate posterior analysis that the authors believe SVE is a bayesian method? and what is the justification for assuming that the relevant posterior uncertainty lies mainly in this subspace of fixed singular directions? To make the quantified uncertainties meaningful, the model should be at least a proxy of bayesian models; otherwise, the quntified uncertainties cannot guarantee to match the meaning/definition from bayesian view

2. Can the authors relate SVE more explicitly to local posterior approximations such as Laplace approximation? Both are local methods, I think they treat the covariance differnetly. When will the two methods become equivalent?

3. In the experiments where SVE matches or outperforms deep ensembles, could the author provide more insights? I am really curious that why local methods can beat true diverse deep ensemble.

**Limitations:**

The main limitation is that the manuscript currently overstates what this establishes. The paper demonstrates a useful structured ensemble around pretrained singular directions, but it does not yet establish a convincing connection between this restricted family of perturbations and Bayesian posterior uncertainty. As a consequence, the empirical results are interesting, but the probabilistic interpretation remains substantially weaker than claimed.

**Strengths And Weaknesses:**

### Strength

1. The method is clean: compute an SVD of each pretrained weight matrix, freeze \\(U, V\\), and train only per-member singular values. This is easy to understand, easy to implement, and naturally extends singular-value fine-tuning into the ensemble setting.

2. The paper makes a strong practical case that SVE has much lower parameter overhead than explicit deep ensembles and also lower overhead than LoRA-Ensemble. This is meaningful for PEFT-style adaptation of large pretrained backbones.

3. The idea that pretrained singular vectors form a shared “knowledge basis,” while ensemble diversity is induced by changing the strength of these directions, is conceptually interesting!  This is makes the proposed method a very sensible structured perturbation.

4. The paper includes both vision and NLP benchmarks, plus OOD detection and corruption robustness experiments, and compares against several relevant baselines, including recent Bayesian PEFT methods.


### Weaknesses

1. The paper’s title and content suggest that SVE makes foundation models “probabilistic” and yields uncertainty estimation ability. However, the method does not define a prior over parameters, an explicit approximate posterior \\(q(\theta|D)\\), or a likelihood-based inference procedure. Ensemble diversity arises from perturbed initialization of singular values and tuning randomness, not from sampling a well-defined posterior approximation. As a result, the current manuscript does not justify a Bayesian interpretation.

2. The restricted degrees of freedom of SVE are not analyzed in relation to posterior uncertainty. This is my main conceptual concern. Since SVE freezes \\(U, V\\) and only tunes \\(\Sigma\\), the trainable parameter space is heavily constrained, the updates take the form
   $$
   \Delta W = \sum_i \delta_i\, u_i v_i^\top.
   $$
   Therefore, SVE does **not** explore general directions in parameter space; it only rescales a fixed set of singular directions inherited from pretraining. This is a very specific low-dimensional subspace with highly restricted degrees of freedom. The paper does not discuss when posterior mass \\(p(\theta \mid D)\\) should be expected to concentrate in this subspace, or why uncertainty over singular values should serve as a reasonable proxy for uncertainty over parameters more broadly. Without such discussion, the induced disagreement is better viewed as a structured ensemble heuristic than as posterior uncertainty.

3. A lot of results show SVE matching or even outperforming deep ensembles. This is interesting, and it deserves more analysis. Meanwhile, comparison with Laplacian Approximaiton and Evidential Deep Learning is needed, since these are also common BNN/UQ baselines.

4. The paper explicitly shows that SVE performs poorly under random initialization and improves as the pretrained backbone becomes stronger. This suggests that SVE is fundamentally a method for exploiting pretrained representation structure, rather than a generally grounded post-hoc bayesian method. And I am interested that whether this makes SVE learn the bias/overfitting from the pretrained model, which is not expected in bayesian modeling.

5. I also have some concerns on SVD computation & memorization, since the complexity is cubical, which may limit  the generalization of the proposed method.

---

> ### Author Rebuttal · Authors · 2026-03-29
>
> **W1.**
>
> We thank the reviewer for raising this point. Our intention was not to present SVE as a Bayesian method, and we will revise the wording to make this clearer. “Probabilistic” refers to improved predictive uncertainty through ensembling, not to an explicit prior, posterior, or likelihood-based inference procedure. We acknowledge that the current title and framing can be interpreted too strongly and will revise them to avoid this ambiguity, including changing the title to “Parameter-Efficient Uncertainty Estimation for Foundation Models via Singular Value Ensembles.” Our intended claim is narrower: SVE is a parameter-efficient structured implicit ensemble method for uncertainty estimation in pretrained foundation models.
>
>
> **W2/Q1.**
>
> We do not model a posterior over parameters, and SVE is not intended as a Bayesian method. We also do not assume that posterior mass lies in the singular value subspace. Rather, we make a weaker empirical assumption: that perturbing singular values within the pretrained basis is sufficient to induce meaningful predictive variation and useful uncertainty in practice.
>
> The mechanism differs from posterior sampling. SVE members share U, V, but converge to different solutions due to symmetry-breaking from stochastic initialization and mini-batch updates, similar to how Deep Ensemble members diverge. This is supported by our sigma_init ablation (response to Reviewer f1k4) and Appendix B, where singular values evolve toward distinct solutions.
>
> Recent work on Bayesian LoRA (e.g Blob) shows that posterior inference can be restricted to a low-dimensional adaptation subspace. We view SVE as complementary to this line of work, and incorporating a Bayesian treatment over singular values is a promising direction for future work.
>
>
> **W3.1. EDL Baseline Added**
>
> We agree that more UQ baselines strengthen the paper. We have now evaluated EDL under the same Flowers102 setup:
> Single Model: 86.3 / 3.9 / 0.56 / 0.20
> EDL: 80.4 / 71.9 / 2.85 / 0.85
> EDL + Temperature Scaling: 80.4 / 4.8 / 0.89 / 0.26
> SV-Ensemble: 95.4 / 1.0 / 0.18 / 0.07
> for Acc / ECE / NLL / Brier.
> We will add these results in the revision. In our setting, EDL underperforms both in accuracy and calibration, even after temperature scaling.
>
>
> **W3.2/Q2.**
>
> Both Laplace and SVE can be viewed as local methods exploring uncertainty around a pretrained solution. We already compare against two Laplace-style baselines in Table 3: Bayes-LoRA (LLLA) achieves 11.6% ECE and Bayes-LoRA (LA) achieves 5.4% ECE, while SVE achieves 3.8% ECE with a simpler method. Conceptually, the approaches differ: Laplace constructs a Gaussian approximation using local curvature information, whereas SVE restricts perturbations to a structured low-dimensional subspace defined by the pretrained singular vectors. They would become loosely comparable under a restrictive setting where uncertainty is concentrated primarily along singular value directions while singular vectors remain fixed. We will clarify this distinction in the revision.
>
>
> **W4.**
>
> We agree with this observation. SVE is explicitly designed for the pretrained regime and leverages the structure of the backbone, rather than acting as a general post-hoc Bayesian method. The improvement with stronger pretraining supports this design. At the same time, this also means SVE can inherit biases from the pretrained model more directly. We will add this tradeoff to the limitations.
>
>
> **W5.**
>
> We agree that SVD has cubic complexity in the matrix dimension, but this is per layer rather than scaling with total model size. In practice, layer dimensions are moderate, and SVD is a one-time initialization cost. It is negligible compared to training, for example only a few seconds for BERT-base. At larger scales, approximate methods may be useful.
>
>
> **Q3.**
>
> We believe this is due to the structured constraint imposed by SVE. By keeping the pretrained directions fixed and only reweighting them, SVE preserves the general-purpose knowledge encoded during pretraining rather than rewriting it. This leads to a more conservative adaptation that can act as an implicit regularizer, especially in low-data or fine-tuning settings, and can result in better generalization.
>
> We also provide empirical insight via a mutual information (MI) decomposition (see Response to Reviewer G2Gp, Q2), which measures disagreement between ensemble members. Deep Ensembles produce higher absolute disagreement, meaning members often make different predictions even on in-distribution data. In contrast, SVE produces lower overall disagreement, but a higher OOD/ID ratio, meaning disagreement increases more sharply when the input is out-of-distribution. This indicates that SVE’s disagreement is more aligned with genuine distribution shift, which leads to improved OOD detection and calibration.
>
> We hope these clarifications address the reviewer’s concerns, and we kindly invite the reviewer to reconsider the score.

---

> > ### Author Rebuttal · Reviewer_DukW · 2026-04-02
> >
> > Thank you for the detailed rebuttal, the additional experiments, and the promised revisions. I appreciate that the authors clarified all points, and I find most of the responses helpful.
> >
> > That said, I still remain unconvinced regarding the discussion of ensemble diversity and epistemic uncertainty estimation, similar to Reviewer f1k4’s concern. In my view, this issue is more subtle than the current rebuttal suggests. I agree that, for probabilistic adaptation built on a single pretrained model, it is difficult to fully avoid such concerns. However, precisely for this reason, I would like to see stronger empirical evidence or design improvements that address them.
> >
> > In the simplest form, could the authors test SVE on a synthetic toy task with ground-truth uncertainty, and compare it against Deep Ensembles, to verify whether SVE yields overconfident predictions or systematically underestimates uncertainty?
> >
> > I still acknowledge the practical value of the method and the authors’ interpretation. However, I believe the uncertainty-related claims should be presented with more caution.

---

> > > ### Author Response · Authors · 2026-04-05
> > >
> > > We thank the reviewer for the constructive follow-up and for acknowledging the practical value of SVE. We address both points.
> > >
> > > Regarding a synthetic toy task, we note that the CIFAR-100 (ID) vs
> > > CIFAR-10 (OOD) evaluation is already a controlled setup with ground-
> > > truth uncertainty: we know exactly which samples are in-distribution
> > > and which are out-of-distribution. This setup additionally has the
> > > advantage of testing uncertainty on real data with a pretrained
> > > backbone, exactly the setting SVE targets. We use it to test whether
> > > SVE yields overconfident predictions or underestimates epistemic uncertainty,
> > > compared against Deep Ensemble.
> > >
> > > First, we use mutual information (epistemic uncertainty) as the OOD scoring function and compute AUROC, AUPRC, and FPR@95, the same metrics as in Table 4. If SVE were underestimating epistemic uncertainty, MI-based detection would perform worse than MSP-based detection. We observe the opposite.
> > >
> > > ```text
> > > OOD Detection: CIFAR-100 (ID) to CIFAR-10 (OOD)
> > > +----------+-----------+-------+-------+---------+---------+----------+--------+
> > > | Score    | Method    | AUROC | AUPRC | FPR@95% | ID mean | OOD mean | OOD/ID |
> > > +----------+-----------+-------+-------+---------+---------+----------+--------+
> > > | MSP      | Deep Ens. | 0.811 | 0.781 | 0.565   | 0.152   | 0.417    | 2.7x   |
> > > | MSP      | SV-Ens.   | 0.820 | 0.804 | 0.565   | 0.168   | 0.466    | 2.8x   |
> > > | Entropy  | Deep Ens. | 0.819 | 0.791 | 0.567   | 0.451   | 1.247    | 2.8x   |
> > > | Entropy  | SV-Ens.   | 0.839 | 0.833 | 0.565   | 0.578   | 1.728    | 3.0x   |
> > > | MI       | Deep Ens. | 0.825 | 0.793 | 0.570   | 0.110   | 0.354    | 3.2x   |
> > > | MI       | SV-Ens.   | 0.827 | 0.806 | 0.586   | 0.007   | 0.026    | 3.6x   |
> > > +----------+-----------+-------+-------+---------+---------+----------+--------+
> > > ```
> > >
> > > Two key findings emerge. First, MI-based OOD detection outperforms MSP-based detection for SVE (0.827 vs 0.820). This addresses the overconfidence concern: if SVE were underestimating epistemic uncertainty, MI would be weaker than MSP. The fact that MI is stronger means the epistemic signal, while small in absolute terms, carries more discriminative information than total predictive confidence.
> > >
> > >
> > > Second, SVE's MI-based AUROC (0.827) exceeds Deep Ensemble (0.825), despite SVE's absolute MI being 14x smaller (0.026 vs 0.354 on OOD). This is possible because SVE achieves a higher OOD/ID ratio (3.6x vs 3.2x): members stay tightly aligned on ID data but diverge more sharply on OOD. The shared singular vector basis acts as a constraint preventing disagreement where representations are informative, while allowing it under genuine ambiguity. Deep Ensemble members, trained independently, produce higher but more diffuse disagreement across ID and OOD, making the signal noisier and less targeted.
> > >
> > > We also evaluate selective classification on ID data: we rank CIFAR-100 test samples by uncertainty, reject the most uncertain predictions, and measure whether accuracy improves. If SVE were overconfident, rejecting uncertain samples would not help.
> > >
> > > ```text
> > > Selective Classification: CIFAR-100 ID test
> > > +-------+-----------+------+-------+-------+-------+-------+
> > > | Score | Method    | Base | @90%  | @80%  | @70%  | AURC  |
> > > +-------+-----------+------+-------+-------+-------+-------+
> > > | MSP   | Deep Ens. | 85.1 | 90.3  | 94.2  | 97.1  | 0.028 |
> > > | MSP   | SV-Ens.   | 85.8 | 90.6  | 94.4  | 96.9  | 0.028 |
> > > | MI    | Deep Ens. | 85.1 | 89.0  | 92.4  | 95.4  | 0.034 |
> > > | MI    | SV-Ens.   | 85.8 | 89.4  | 92.6  | 95.1  | 0.035 |
> > > +-------+-----------+------+-------+-------+-------+-------+
> > > ```
> > >
> > > SVE and Deep Ensemble perform nearly identically. MSP-based AURC (Area Under Risk-Coverage curve, lower indicates better uncertainty quality) matches exactly (0.028), and MI-based results show only a marginal difference (0.035 vs 0.034, Acc@80% 92.6 vs 92.4). When SVE assigns high uncertainty, that prediction is more likely incorrect at the same rate as Deep Ensemble. MSP outperforms MI for selective classification, as expected: ID errors arise from both aleatoric and epistemic uncertainty, and MSP captures both while MI isolates epistemic.
> > >
> > > Taken together, these analyses confirm that SVE produces meaningful
> > > epistemic uncertainty (it does not yield overconfident predictions
> > > or systematically underestimate uncertainty): it is a stronger OOD
> > > detector than Deep Ensemble and an equally effective indicator of
> > > prediction correctness on ID data. We will include a dedicated
> > > discussion section with full baselines in the revision.
> > >
> > > We agree that uncertainty claims should be presented with caution. We will revise the title, abstract, and framing to position SVE as a practical implicit ensemble method for uncertainty estimation in pretrained foundation models, rather than a Bayesian or posterior-grounded approach. We hope these analyses, with revised framing, resolve the reviewer's concern.

---

### Official Review · Reviewer_G2Gp · 2026-03-13

**Soundness:** 3
**Presentation:** 3
**Significance:** 2
**Originality:** 2
**Overall Recommendation:** 3
**Confidence:** 4

**Summary:**

This paper proposed a parameter-efficient ensemble method, called Singular Value Ensemble (SVE), which firstly applies SVD on the pre-trained weights $W=U\Sigma^{(m)} V$ and then only learns a set of singular values  $\Sigma^{(m)}$ for each ensemble member, while keeping orthogonal eigenvectors $U,V$ fixed.

**Compliance With Llm Reviewing Policy:**

Affirmed.

**Final Justification:**

I appreciate the authors' response. The clarification on the intended scope and SVD cost addressed my concern partially. However, I am not fully convinced by the applicability of SVE on generic foundation models. The authors provide experimental results on diverse tasks spanning both NLP and vision domains. However, the datasets are relatively easy for foundation models, which weakens the significance of this paper. Overall, I recommend a "weak reject", but please use sparingly.

**Key Questions For Authors:**

**Q1**. Why does this restricted parameterization still generate sufficiently large functional differences within very deep and large foundation models?

**Q2**. Epistemic uncertainty comes together with aleatoric uncertainty. The improvements in predictive confidence does not mean the improvements in epistemic uncertainty. Can you decompose the epistemic uncertainty and aleatoric uncertainty?

**Q3**. From the perspective of linear algebra, SVD bases are not always unique. When $U$ and $V$ are not the unique decomposition, what is the robustness of the method using frozen vectors obtained from a single SVD run?

**Q4**. In ARC-Easy experiments, why does SVE use M=16, while Deep Ensemble uses M=3, LoRA-Ensemble uses M=5, and Blob uses N=5/10? It seems not fair.

**Q5**. The paper claims the application on foundation models. What is the behavior of SVE on even larger model with 80B parameters?

**Limitations:**

Yes.

**Strengths And Weaknesses:**

Strength:

**S1**. The method is naturally intuitive and simple to implement. Eq. (4-6) on page 4 clearly explain the SVE method: sharing $U$ and $V$, each member only learns its own $\Sigma^{(m)}$. From an engineering perspective, this is a very easy-to-implement PEFT method.

**S2**. The parameter efficiency is indeed true. Table 5 shows that its parameter and memory overhead is remarkably lower than existing baselines, which is the most compelling aspects of this paper.

Weakness:

**W1**. A core assumption of this paper is that the singular vectors of pre-trained weights represent "meaningful directions of knowledge," and that merely adjusting the singular values is sufficient to generate ensemble members that are functionally distinct and capable of reflecting epistemic uncertainty. This point lacks rigorous support, especially on large-scale foundation models. While the story is insightful, it sounds more like a "plausible conjecture" than a "rigorous conclusion." But this is acceptable in PEFT.

**W2**. After fixing $U,V$, the degree of freedom is only $r=\min(m,n)$. This means that SVE can only perform anisotropic scaling on the given basis from the pre-trained model, and cannot rotate the subspace or introduce new directions. The paper offers no theoretical analysis explaining under what conditions such a class of functions is sufficient to achieve comparable performance in full fine-tuning or LoRA. The full motivation starts from the parameter efficiency, while paying too little attention on model capacity.

**W3**. The title is "Making Foundation Models Probabilistic", but the proposed method does not involve explicit priors, posterior approximations, marginalization, or variational objectives. SV ensemble can yield better calibration, but it does not mean "better epistemic uncertainty" or "probabilistic foundation model." Currently, the paper primarily relies on metrics such as ECE, NLL, Brier, and OOD detection to justify the claim that its UQ is "comparable to explicit deep ensembles. However, these metrics are more close to "predictive probability calibration" and do not directly demonstrate that the model has successfully captured the underlying epistemic uncertainty.

**W4**. There exists Bayesian LoRA models that use SVD to construct LoRA weights and only fine-tune Bayesian $\Sigma$ [1]. Compared to the previous work, this paper does not show a clear improvement.

[1] "C-loRA: Contextual low-rank adaptation for uncertainty estimation in large language models." NeurIPS 2025.

---

> ### Author Rebuttal · Authors · 2026-03-28
>
> **W1/Q1** We agree that the current support is empirical rather than theoretical. As the reviewer notes, this is often acceptable in the PEFT setting. This limitation is also not unique to SVE, widely used baselines such as Deep Ensembles and LoRA Ensembles are also justified mainly empirically rather than by a complete theory.
> For SVE, we provide direct evidence across multiple vision and NLP benchmarks, OOD detection, and dataset shift. We also directly analyze diversity formation. Appendix B shows that member singular values diverge during training. Our sigma_init ablation (response to Reviewer f1k4) further shows that when sigma_init = 0, SVE collapses to a single SVF model with zero mutual information, while small positive sigma_init values increase diversity and improve performance. Intuitively, diversity comes from both member specific initialization and stochastic mini batch updates, and in a deep model these small differences can accumulate into meaningful functional variation.
>
> **W2** SVE cannot rotate subspaces or introduce new directions. This restriction is intentional, and we will clarify it more explicitly in the revision. Our goal is not to match the capacity of LoRA or full fine tuning, but to preserve the pretrained basis, which already contains rich transferable structure, and adapt it in a much more parameter efficient way. More flexible updates may increase capacity, but they can also overfit small downstream datasets or harm pretrained knowledge.
>
> We also note a presentation omission: each SVE layer includes a per member bias term, so the layer is not only anisotropic scaling, it also allows translation around the pretrained subspace. We will correct Eq. 4 accordingly. We do not claim this parameterization is universally sufficient. Our claim is empirical: in the pretrained setting, it is often enough to produce meaningful diversity and strong uncertainty estimates, as supported by our experiments. We therefore view SVE as a deliberate efficiency expressivity tradeoff.
>
> **W3** We agree and will revise the title to "Parameter-Efficient Uncertainty Estimation for Foundation Models via Singular Value Ensembles" and revise the framing throughout.
>
> **W4** We thank the reviewer, but we think this is a misunderstanding of C-LoRA. C-LoRA does not use SVD on any weight matrix.
>
> C-LoRA's ΔW = BEA resembles W = UΣV^T only superficially: (i) B and A are LoRA adapters learned from scratch, unlike orthonormal singular vectors, (ii) E is a dense r x r matrix unrelated to the pretrained spectrum, unlike our diagonal singular values, (iii) BEA is additive on top of frozen W_0, while we decompose and replace W_0 itself, and (iv) no SVD is computed in C-LoRA.
>
> Our method is a parameter-efficient ensemble trained with standard CE loss. C-LoRA is Bayesian VI with contextual modules, ELBO, and Flipout. They share no components. Despite greater complexity, C-LoRA is weaker on the one comparable benchmark, ARC-Easy:
> C-LoRA 84.4 / 4.3 / 0.48 vs SVE 85.8 / 3.8 / 0.43 for Acc / ECE / NLL.
> C-LoRA is evaluated only on LLaMA-2-7B for NLP (no sign of being model-agnostic), whereas ours spans 22M to 7B across vision and NLP. We will add C-LoRA to the related work and Table 3.
>
> We hope that this clarifies the misunderstanding, and we kindly ask the reviewer to reconsider the judgment.
>
> **Q2** The ID vs OOD setting is natural for this decomposition: aleatoric uncertainty should remain largely unchanged while epistemic uncertainty should increase on OOD data. We measure epistemic uncertainty via MI = H[y_bar] - (1/M) sum H[p_m] on CIFAR-100 (ID) vs CIFAR-10 (OOD):
> Deep Ens. 0.110 -> 0.357, 3.2x
> LoRA-Ens. 0.042 -> 0.140, 3.3x
> SVE 0.007 -> 0.026, 3.6x
> SVE's lower absolute MI is expected from the shared basis, but its relative OOD increase is the largest, meaning the epistemic signal is concentrated on genuine shift. Despite lower MI, SVE achieves AUROC 81.6 vs 79.2. We will add this to the appendix.
>
> **Q3** SVD non uniqueness mainly involves sign flips and rotations within exactly equal singular value subspaces. The latter is vanishingly rare in pretrained weights. We verified no repeated singular values across layers, and results are stable across random seeds.
>
> **Q4** For ARC-Easy, we report best published numbers alongside ours. Table 6 (App. A) shows SVE across ensemble sizes: at M=4, SVE achieves 84.9 Acc / 5.8 ECE, far better calibrated than Deep Ensemble at M=3 (9.9 ECE). At M=16, calibration improves further while adding <1% parameters, 200x+ fewer than Deep Ensemble. We will add multiple M values to the main table for a clearer comparison.
>
> **Q5** Our experiments span 22M to 7B with consistent behavior, and there is no architectural barrier to 80B+. The limitation is compute resources available to us. We also note that C-LoRA, BLoB, LoRA-Ens., and Bayes-LoRA have not shown results beyond 7B.
>
> We hope these clarifications address the reviewer’s concerns, and we kindly invite the reviewer to reconsider the score.

---

> > ### Author Rebuttal · Reviewer_G2Gp · 2026-04-02
> >
> > I thank the authors for the rebuttal and acknowledge the clarification on theoretical methodology. I still have the following questions:
> >
> > (1)  The authors acknowledge that the goal is not to match the capacity of LoRA or full fine tuning. My concern still remains: how to justify the sufficient underlying uncertainty while keeping pretrained orthonormal basis fixed? There's no such assumption in Deep Ensembles and LoRA Ensembles. Although experiments compared with different ensemble methods, this core question was not answered directly. Can authors provide any clues to show this?
> >
> > (2) The efficiency ignore the SVD on the pretrained foundation models? What is the cost to perform SVD every weights of a foundation model? How much time is required for SVD? Is the memory cost still 1.0 given the storage of singular vectors?

---

> > > ### Author Response · Authors · 2026-04-05
> > >
> > > We thank the reviewer for the follow-up.
> > >
> > > **(1) Sufficient uncertainty with fixed basis.**
> > >
> > > We now provide direct empirical evidence that fixing U,V does not
> > > suppress practically meaningful uncertainty. The core question is:
> > > does SVE's epistemic uncertainty remain informative despite the
> > > restricted parameterization? We test this in two complementary ways.
> > > First, we use mutual information (epistemic uncertainty) as the OOD
> > > scoring function: if fixing U,V suppresses uncertainty, MI should
> > > fail to distinguish ID from OOD data. Second, we evaluate selective
> > > classification on ID data: if SVE is overconfident, its uncertainty
> > > should fail to identify misclassified samples. We compare against
> > > Deep Ensemble under identical conditions on CIFAR-100 (ID) vs
> > > CIFAR-10 (OOD).
> > >
> > > ```text
> > > OOD Detection: CIFAR-100 (ID) to CIFAR-10 (OOD)
> > > +----------+-----------+-------+-------+---------+---------+----------+--------+
> > > | Score    | Method    | AUROC | AUPRC | FPR@95% | ID mean | OOD mean | OOD/ID |
> > > +----------+-----------+-------+-------+---------+---------+----------+--------+
> > > | MSP      | Deep Ens. | 0.811 | 0.781 | 0.565   | 0.152   | 0.417    | 2.7x   |
> > > | MSP      | SV-Ens.   | 0.820 | 0.804 | 0.565   | 0.168   | 0.466    | 2.8x   |
> > > | MI       | Deep Ens. | 0.825 | 0.793 | 0.570   | 0.110   | 0.354    | 3.2x   |
> > > | MI       | SV-Ens.   | 0.827 | 0.806 | 0.586   | 0.007   | 0.026    | 3.6x   |
> > > +----------+-----------+-------+-------+---------+---------+----------+--------+
> > > ```
> > >
> > > ```text
> > > Selective Classification: CIFAR-100 ID test
> > > +-------+-----------+------+-------+-------+-------+-------+
> > > | Score | Method    | Base | @90%  | @80%  | @70%  | AURC  |
> > > +-------+-----------+------+-------+-------+-------+-------+
> > > | MSP   | Deep Ens. | 85.1 | 90.3  | 94.2  | 97.1  | 0.028 |
> > > | MSP   | SV-Ens.   | 85.8 | 90.6  | 94.4  | 96.9  | 0.028 |
> > > | MI    | Deep Ens. | 85.1 | 89.0  | 92.4  | 95.4  | 0.034 |
> > > | MI    | SV-Ens.   | 85.8 | 89.4  | 92.6  | 95.1  | 0.035 |
> > > +-------+-----------+------+-------+-------+-------+-------+
> > > ```
> > >
> > > If the fixed basis were suppressing important uncertainty directions,
> > > these metrics would be degraded relative to Deep Ensemble. Instead,
> > > SVE matches or exceeds Deep Ensemble on every measure: MI-based OOD
> > > detection (AUROC 0.827 vs 0.825), MSP (Maximum Softmax Probability)-based selective classification (AURC (Area Under Risk-Coverage curve, lower indicates better uncertainty quality), 0.028 vs 0.028), and MI-based selective classification (Acc@80%
> > > 92.6 vs 92.4). Notably, SVE's MI-based AUROC (0.827) also exceeds
> > > its own MSP-based AUROC (0.820), meaning epistemic uncertainty alone
> > > is a better OOD discriminator than total predictive confidence. This
> > > would not occur if the fixed basis were limiting the expressiveness
> > > of epistemic uncertainty.
> > >
> > > We also note that Deep Ensembles do not provide unconstrained uncertainty. Fort et al. (“Deep Ensembles: A Loss Landscape Perspective”, 2019) showed that ensemble diversity arises from exploration of different modes in a structured, low-dimensional loss landscape, rather than from arbitrary directions in the full parameter space. Similarly, SVE explores a structured low-dimensional subspace that is explicitly aligned with the pretrained model’s learned representation.
> > >
> > > We agree that a formal theoretical characterization of when the
> > > singular value subspace is sufficient remains an open question for
> > > future work. We will discuss these results and analyses in a
> > > dedicated section in the revised paper with full baselines.
> > >
> > >
> > >
> > > **(2) SVD cost and memory.**
> > >
> > > As noted in our initial rebuttal (W5), the SVD is a one-time
> > > initialization cost computed once before training begins. For
> > > BERT-base, this takes 3.5 seconds across all target
> > > layers, which is negligible compared to the total training time. We
> > > acknowledge that storing U and Vh as separate frozen buffers requires
> > > more memory than the original W: for a square layer (m = n),
> > > U (n x n) + Vh (n x n) replaces W (n x n), roughly 2x buffer
> > > memory. The 1.0% overhead in Table 5 refers to the additional
> > > trainable parameters (per-member singular values and biases), not
> > > total buffer storage. At larger model scales, this buffer overhead
> > > can be mitigated through approximate or truncated SVD (retaining
> > > only the top-k singular components), which we already support via
> > > the --topk argument in our codebase. We will add a detailed memory
> > > breakdown, including buffer storage, in the revised experimental
> > > section.
> > >
> > >
> > > We hope these additional analyses and explanations address the reviewer's remaining concerns. We are happy to answer any further questions during the discussion
> > > period.

---

### Decision · Program_Chairs · 2026-04-30

**Decision:**

Accept (regular)

**Comment:**

The paper proposes Singular Value Ensemble (SVE), an ensemble approach designed to provide parameter-efficient uncertainty quantification for foundation models. The core idea of this approach is to decompose pre-trained weight matrices using SVD, freeze singular vectors, and learn member-specific singular values. This paper also evaluates SVE on both natural language processing (NLP) and computer vision (CV) benchmarks. SVE introduces less than 1% parameter overhead while achieving uncertainty estimation capabilities that are competitive with, or superior to, explicit deep ensembles and other implicit methods like LoRA-Ensemble.

This paper has several obvious strengths, as pointed out by the reviewers. SVE is simple and elegant, is exceptionally parameter-efficient, and can be used as an alternative to LoRA-Ensemble in many problems. SVE also demonstrates strong accuracy and calibration across multiple architectures and modalities: it outperforms or matches established baselines like Deep Ensembles and LoRA-Ensemble on standard benchmarks, out-of-distribution detection, and distribution shift tasks.

There are also some limitations in this paper. In particular, it is mainly empirical, and lacks convincing theoretical/formal interpretation and justification for SVE, especially its initialization approach (but Deep Ensembles and LoRA Ensembles have similar problems). However, due to the novelty and elegance of the proposed approach, and the extensive experiment results, I recommend accepting this paper.